# 🦉 Grasp Any Region: Towards Precise, Contextual Pixel Understanding for Multimodal LLMs

**Haochen Wang**[1,2*] **Yuhao Wang**[3*] **Tao Zhang**[4] **Yikang Zhou**[4] **Yanwei Li**[5] **Jiacong Wang**[2]
**Jiani Zheng**[5] **Ye Tian**[3] **Jiahao Meng**[3] **Zilong Huang**[5] **Guangcan Mai**[5] **Anran Wang**[5]
**Yunhai Tong**[3] **Zhuochen Wang**[5] **Xiangtai Li**[5†] **Zhaoxiang Zhang**[1,2†]

[1]New Laboratory of Pattern Recognition (NLPR),
 State Key Laboratory of Multimodal Artificial Intelligence Systems (MAIS),
 Institute of Automation, Chinese Academy of Sciences (CASIA)
[2]University of Chinese Academy of Sciences  [3]Peking University  [4]Wuhan University  [5]ByteDance
 {wanghaochen2022, zhaoxiang.zhang}@ia.ac.cn   xiangtai94@gmail.com

## Abstract

While Multimodal Large Language Models (MLLMs) excel at *holistic* understanding, they struggle in capturing the dense world with complex scenes, requiring fine-grained analysis of intricate details and object inter-relationships. Region-level MLLMs have been a promising step. However, previous attempts are generally optimized to understand given regions *in isolation*, neglecting crucial global contexts. To address this, we introduce **G**rasp **A**ny **R**egion (**GAR**) for *comprehensive* region-level visual understanding. Empowered by an effective RoI-aligned feature replay technique, **GAR** supports (1) precise perception by leveraging necessary global contexts, and (2) modeling interactions between multiple prompts. Together, it then naturally achieves (3) advanced compositional reasoning to answer specific free-form questions about any region, shifting the paradigm from passive description to active dialogue. Moreover, we construct **GAR-Bench**, which not only provides a more accurate evaluation of single-region comprehension, but also, more importantly, measures interactions and complex reasoning across *multiple regions*. Extensive experiments have demonstrated that **GAR-1B** not only maintains the state-of-the-art captioning capabilities, *e.g.*, outperforming DAM-3B +4.5 on DLC-Bench, but also excels at modeling relationships between multiple prompts with advanced comprehension capabilities, even surpassing InternVL3-78B on GAR-Bench-VQA. More importantly, our *zero-shot* **GAR-8B** even outperforms in-domain VideoRefer-7B on VideoRefer-Bench[Q], indicating its strong capabilities can be easily transferred to videos. The code is available at https://github.com/Haochen-Wang409/Grasp-Any-Region.

## 1 Introduction

The ambition of Multimodal Large Language Models (MLLMs) is to endow machines with human-like abilities to perceive, interpret, and reason about the *dense* visual world (Yuan et al., 2024; Lian et al., 2025; Li et al., 2025). To date, renowned state-of-the-art models (Bai et al., 2025; Wu et al., 2024; Wang et al., 2025e; DeepMind, 2025b; OpenAI, 2024a;b; 2025) have made remarkable strides, excel in generating *holistic* descriptions and answering general questions about an *entire* image. However, this global-level perception struggles with the *dense* understanding of cluttered environments, intricate object details, and the complex interplay between multiple entities.

To address the limitation of global perception, several previous works (Chen et al., 2023; Yuan et al., 2024; Zhang et al., 2024a; Lian et al., 2025; Lin et al., 2025b) argue for a paradigm shift to region-level MLLMs. Specifically, they equip MLLMs with *promptable* and *fine-grained* interactions

---

*Equal contribution. †Corresponding authors.

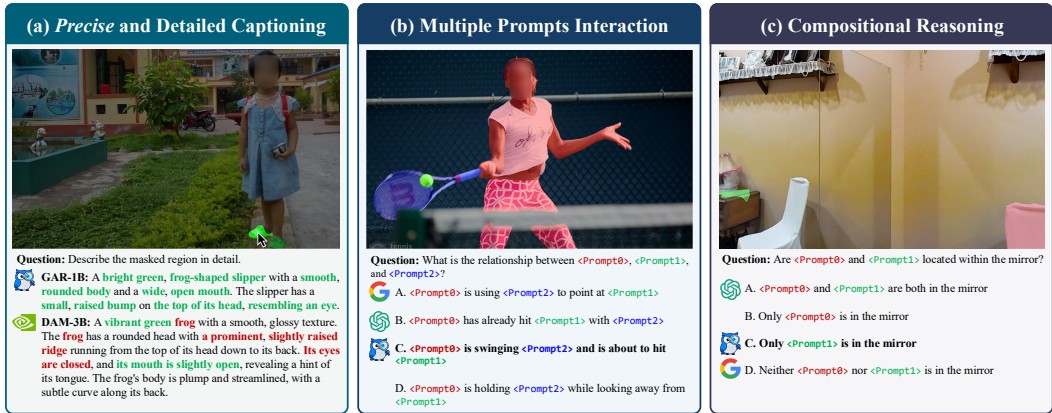

Figure 1: **Illustration of our GAR**, which is superior at *leveraging necessary global context* to (a) generate *precise* captions, where green is correct and red means wrong, (b) model complex interactions among *multiple prompts*, and perform reasoning such as (c) recognizing non-entities. Colors of `<Prompt0>`, `<Prompt1>`, and `<Prompt2>` correspond to masks with respective colors. Image source: (a) Shao et al. (2019), (b) Lin et al. (2014), and (c) Mei et al. (2021).

to achieve targeted region-level understanding, using boxes (Zhang et al., 2024a; Chen et al., 2023) or masks (Yuan et al., 2024; Lian et al., 2025). This mechanism transforms the model from a passive observer of the entire scene into an active participant capable of deep, localized analysis. However, effectively balancing global scene context with fine-grained local details remains challenging, which serves as a fundamental trade-off in region-level MLLMs. Conventional methods (Yuan et al., 2024; You et al., 2023) that employ pooled local features suffer from insufficient details, while recent models (Lian et al., 2025; Lin et al., 2025b) mainly focus on the ability to generate a descriptive *caption* for a *single* region, and thus model architectures are generally optimized to understand a given region *in isolation*. This design often neglects crucial global context, *e.g.*, misidentifying a frog-shaped slipper as a real frog in Figure 1a. To this end, we propose **G**rasp **A**ny **R**egion (**GAR**) for *comprehensive and detailed* region understanding. As shown in Figure 1, key features include:

**(1) Precise Perception.** Thanks to the leverage of necessary global contexts, GAR achieves a more precise perception of given regions, which is the fundamental capability for region MLLMs. As shown in Figure 1a by aggregating information from the broader, *unmasked* scene, our GAR manages to generate much more accurate descriptions than previous crop-based approaches (Lian et al., 2025).

**(2) Interactions between Multiple Prompts.** GAR moves beyond the prevailing single-prompt paradigm (Lian et al., 2025), which treats every region of interest as an *isolated entity*. As illustrated in Figures 1b and 1c, GAR manages to model relationships between an arbitrary number of prompts.

**(3) Advanced Compositional Reasoning Capabilities.** Empowered with the aforementioned features, GAR is naturally equipped with advanced compositional reasoning capabilities, allowing it to answer any specific free-form questions.

To achieve these capabilities, effectively encoding global contexts becomes equally crucial as local detailed features. To this end, we propose an RoI-aligned feature replay technique. Specifically, **GAR** first encodes the full, uncropped image (together with the mask prompt) with AnyRes (Liu et al., 2024). Subsequently, RoI-Align (He et al., 2017) is employed to gather relevant features directly from the global feature map. Those gathered features are *inherently context-aware*, providing sufficient local details while maintaining global information simultaneously. Please refer to Figure 3 for the detailed pipeline.

Furthermore, we introduce **GAR-Bench**, which not only provides a more accurate evaluation of single-region comprehension by constructing multiple-choice questions, but also, more importantly, measures *interaction* and *complex reasoning* across *multiple regions*. It includes test cases that require a

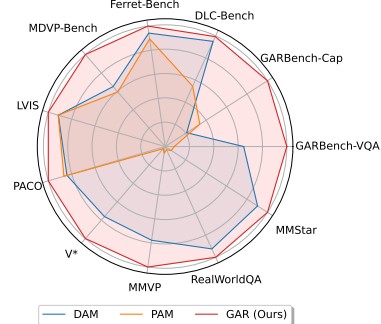

Figure 2: **Performance comparison. GAR** achieves strong performances not only on region-level understanding, but also excels in general multimodal benchmarks.

model to aggregate information from multiple visual regions to arrive at a correct conclusion, thereby quantifying the ability to interpret the whole scene rather than independent parts.

Empirically, shown in Figure 2, our **GAR-1B** not only outperforms DAM-3B (Lian et al., 2025) and PAM-3B (Lin et al., 2025b) on detailed captioning benchmarks (Lian et al., 2025; You et al., 2023; Lin et al., 2025a), but also excels in general multimodal benchmarks (Wu & Xie, 2024; Tong et al., 2024b; xAI, 2024; Chen et al., 2024a). Interestingly, it even outperforms large-scale models like InternVL3-78B (Zhu et al., 2025) on **GAR-Bench**, demonstrating its advanced comprehension capability in modeling interactions between multiple prompts. More importantly, our *zero-shot* **GAR-8B** even outperforms in-domain VideoRefer-7B on VideoRefer-Bench$^Q$, indicating its strong comprehension capabilities can be easily transferred to videos. We hope our work inspires the community to develop MLLMs that can perceive and understand the dense visual world more effectively.

## 2 RELATED WORKS

**Multimodal Large Language Models (MLLMs).** Typical MLLMs (Liu et al., 2023; Li et al., 2024; Liu et al., 2024; Bai et al., 2025; Zhu et al., 2025; Wu et al., 2024; Lei et al., 2025; Wang et al., 2025b;c; Tong et al., 2024a; Yang et al., 2024a;b; 2025b;a; 2026) project visual features extracted from pre-trained visual encoders (Radford et al., 2021; Zhai et al., 2023) to LLM for understanding multimodal contents. However, these models usually lack precise localization capabilities (Lian et al., 2025; Lin et al., 2025b) and struggle to understand specific regions. One potential solution is to "think with images" (OpenAI, 2025; Wang et al., 2025d;a). But these agentic models require complex multi-turn conversations, while we mainly focus on precise perception within a single-turn dialogue.

**Region-Level MLLMs.** Different from conventional image-level comprehension, localized understanding requires MLLMs to capture regional attributes. Previous methods either utilize visual markers (Yang et al., 2023), bounding boxes (Zhang et al., 2024a; Chen et al., 2023; You et al., 2023; Rasheed et al., 2024; Lee et al., 2024; Ma et al., 2024), or segmentation masks (Yuan et al., 2024; Lian et al., 2025), to represent regions-of-interests within an image. We simply regard masks as visual prompts, since masks have less ambiguity than other representations. Beyond prompt representations, effectively balancing global scene context with local details remains an open problem. Existing methods struggle to master both. Methods like DAM (Lian et al., 2025) excel at local perception but lack a holistic view of the global context. Conversely, earlier works like GPT4RoI (Zhang et al., 2024a) and GLaMM (Rasheed et al., 2024), while incorporating the global image, tend to lose crucial local details by pooling region features into single vectors. **GAR** is designed specifically to solve this dilemma. Based on this, **GAR** excels in modeling the relationship between an arbitrary number of visual prompts while effectively maintaining crucial global context and sufficient local details.

**Benchmarks for Region-Level Understanding.** Typical region-level benchmarks only evaluate the *caption* quality for *single prompt* using conventional language-based captioning metrics (You et al., 2023; Yuan et al., 2024; Zhang et al., 2024b; Guo et al., 2024; Rasheed et al., 2024), model-based similarities (Chen et al., 2025; Yuan et al., 2024), and LLM-Judged accuracies without the need for reference captions (Lian et al., 2025). **GAR-Bench** is to systematically evaluate the comprehension capabilities with multiple visual prompts. It contains a caption protocol to measure the correctness of descriptions for the relation between visual prompts, and a VQA protocol to evaluate *both* the basic understanding capability for specific regions, *e.g.*, color and shape, and advanced compositional reasoning abilities for multiple regions.

## 3 GRASP ANY REGION

We start from the *task formulation* in Section 3.1. Subsequently, we introduce our *model architecture* and *training data pipeline* in Section 3.2 and Section 3.3, respectively. Finally, we introduce our *benchmark designs* in Section 3.4 to systematically evaluate region-level comprehension capabilities.

### 3.1 TASK FORMULATION

The task of grasping any region is a hierarchical challenge from basic perception to complex, compositional reasoning about specific visual regions. Specifically, given an image $I \in \mathbb{R}^{H \times W \times 3}$, where $H \times W$ indicates the resolution, and a *set* of $N$ binary visual prompts, *e.g.*, masks $\{M_i\}_{i=1}^N$, where $M_i \in \{0, 1\}^{H \times W}$, the objective is to generate a precise text response $R$ that demonstrates a

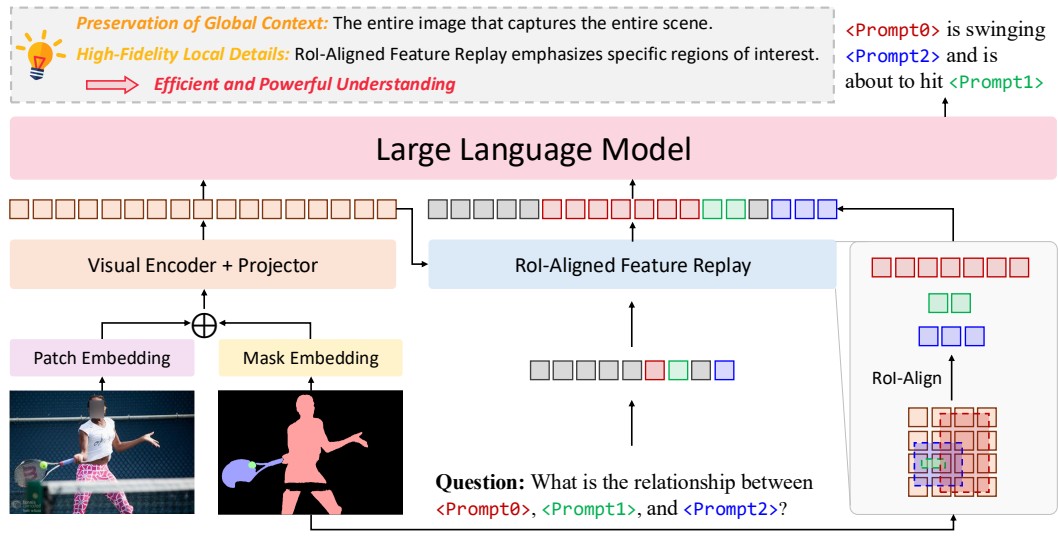

Figure 3: **Illustration of our GAR.** It leverages a single-pass visual encoder to create a holistic feature map of the entire scene, thus preserving global context. Simultaneously, an "RoI-Aligned Feature Replay" mechanism extracts high-fidelity features for specific objects of interest. Both the global context features and the detailed local features are then fed into an LLM to accurately infer complex relationships and interactions between multiple objects within the image.

multi-layered comprehension of the scene, *e.g.*, detailed attributes description and relational caption, based on the given text instruction $T$:

$$R = \text{RegionModel}\left(\boldsymbol{I}, \{\boldsymbol{M}_i\}_{i=1}^N, T\right). \tag{1}$$

Specifically, this task is structured in three ascending levels of capability: *(1)* Generating detailed descriptions for a *single* region is the foundation, *e.g.*, "describe <Prompt1> in detail", where <Prompt1> actually denotes a binary mask and is specified by the user. It requires the model to accurately perceive and articulate the fine-grained attributes contained strictly within the boundaries of a given prompt. *(2)* The next stage requires understanding the given region with the necessary global contexts. This moves beyond isolated analysis, requesting to aggregate information from the broader, *unmasked* scene. This capability is critical for advanced reasoning tasks such as *position identification* (*i.e.*, locating an object as "the second from the left in the third row") and *non-entity recognition* (*e.g.*, correctly identifying a reflection in a mirror versus a physical object), where the prompt itself is insufficient for a correct interpretation. *(3)* Finally, the task culminates in the ability to perceive, understand, and describe the relationship between *multiple* regions. This assesses the capacity for true compositional reasoning by requiring it to articulate the spatial, functional, or interactive connections between *different prompts*.

## 3.2 MODEL ARCHITECTURE

The task definition above requires overcoming the contextual blindness inherent in models that analyze prompted regions *in isolation*. As established, this myopic focus can lead to fundamental reasoning errors, such as misidentifying a frog-shaped slipper as a real frog because the surrounding bedroom context is ignored. Therefore, our architectural design of Grasp Any Region (**GAR**) is guided by a central principle: to achieve *a fine-grained understanding of the prompted region while simultaneously preserving and leveraging the global context of the entire scene*. Illustrated in Figure 3, we introduce two new components into the architecture: (1) a simple yet effective prompt encoding scheme, and (2) a novel RoI-aligned feature replay technique.

**Prompt Encoding and Integration.** To integrate spatial guidance into the vision backbone, we introduce a lightweight prompt encoding mechanism similar to Lian et al. (2025) and Sun et al. (2024). The input binary mask, which specifies the region(s) of interest, is first processed by a simple convolutional block (LeCun et al., 1989) to produce a mask embedding. This zero-initialized (Zhang et al., 2023) mask embedding is then added to ViT's (Dosovitskiy et al., 2021) patch embeddings.

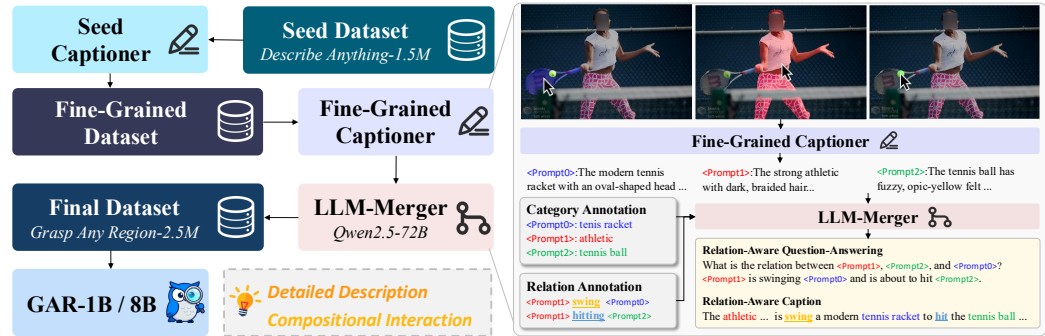

Figure 4: **Illustration of our training data pipeline**, which mainly includes two rounds of captioning and judging. Specifically, (1) starting from using the seed dataset to train a seed captioner, we first construct 456K fine-grained descriptions. Subsequently, (2) we utilize both datasets to obtain a fine-grained captioner, and leverage the annotations of the Panoptic Scene Graph (PSG) dataset (Yang et al., 2022) to provide sufficient relation-aware captions and question-answering pairs. Finally, our GAR models are trained with all three parts.

**RoI-aligned Feature Replay.** To simultaneously provide sufficient local details and maintain necessary global context, we introduce the RoI-aligned feature replay technique. Specifically, our model processes the full, uncropped image (with the encoded mask prompt) with AnyRes (Liu et al., 2024), producing a global feature map that is rich in contextual information. Based on the input mask, we then derive a corresponding bounding box for the region of interest and employ RoI-Align (He et al., 2017) to gather the relevant feature vectors directly from the global feature map. Because the features are extracted from a feature map that was computed over the entire image, *they are inherently context-aware*, which elegantly avoids the pitfalls of local-only processing in Lian et al. (2025). At the same time, it provides the subsequent language model with a sufficiently detailed, high-resolution representation of the prompted region, enabling it to perform fine-grained understanding. This replay of context-rich features allows **GAR** to simultaneously "zoom in" on detail without "losing sight" of the bigger picture. Ablations of this design can be found in Table 8, where we demonstrate that this design is capable of both (1) providing sufficient local details and (2) preserving global contexts.

## 3.3 TRAINING DATA PIPELINE

To enhance model capabilities from basic object recognition with *single* region to complex relational reasoning with *multiple* regions, we design a multi-stage process to generate a large-scale, high-quality dataset, as illustrated in Figure 4. Ablations of each round can be found in Table 10. Prompts for each stage can be found in Appendix G.

**Round 1: Enhance Recognition Capability.** Initially, we start from the Describe Anything-1.5M dataset (Lian et al., 2025). However, we observe deficiencies in its fine-grained recognition capability, limiting the quality of generated captions for more complex scenarios. To address this, we integrated images and masks provided by Sun et al. (2024), which is a subset of ImageNet-21K (Deng et al., 2009), an extremely fine-grained classification dataset and renowned for its detailed and extensive category labels. We employ the seed captioner to generate descriptions and then utilize an LLM to validate these generated captions against the ground-truth categories, resulting in a refined fine-grained dataset of 456K samples. We utilize both datasets to train a fine-grained captioner.

**Round 2: Supporting Multiple Prompts.** To further enable understanding multiple prompts, we incorporated the Panoptic Scene Graph (PSG) dataset (Yang et al., 2022), which is rich in relational information. We first query the fine-grained captioner to generate a detailed description for each region. Subsequently, we regard Qwen2.5-72B (Team, 2024) as the LLM-Merger, together with the original annotations provided by the PSG dataset (Yang et al., 2022), to generate: (1) 144K rich object descriptions that explicitly integrate relational context, (2) 144K question-answering pairs designed to probe the understanding of complex relationships, and (3) 126K multiple-choice questions. We construct a relation dataset with 414K samples in total during this stage.

## 3.4 GAR-BENCH

Finally, we introduce **GAR-Bench**, a comprehensive benchmark suite designed to systematically evaluate the region-level comprehension capabilities of MLLMs *beyond simply describing a single region*. Specifically, it is structured into two primary components: a multi-prompt captioning task (**GAR-Bench-Cap**) and a multifaceted visual question answering task (**GAR-Bench-VQA**). The captioning component is designed to assess a model's ability to describe the complex relationships and interactions between multiple visual prompts in a cohesive narrative. The VQA component further dissects a model's understanding into two key areas: (1) its ability to perceive basic attributes for a given prompt, and (2) its capacity for advanced, region-centric compositional reasoning that requires synthesizing information from the prompt and its surrounding context.

**GAR-Bench-Cap** goes beyond isolated object descriptions and measures the ability to perform *compositional scene understanding*. In this task, a model is provided with an image and *two or more* distinct visual prompts. It contains two sub-tasks: (1) simply describe the relationship, and (2) generate detailed captions including necessary relationships. For the "*simple*" protocol, models are directly asked with "what is the relationship between `<Prompt1>` and `<Prompt2>`" and are required to answer the question simply. For the "*detailed*" protocol, for instance, `<Prompt1>` highlights a person and `<Prompt2>` is a bike, the model is not evaluated on its ability to describe each independently, but rather on its capacity to generate an accurate description of their relation like, "`<Prompt1>` is riding `<Prompt2>`". The models need to perform spatial reasoning, action recognition, and semantic integration across disparate image regions, thereby quantifying its ability to interpret a scene as a cohesive whole rather than a collection of independent parts.

**GAR-Bench-VQA** is designed to shift the evaluation from static description to dynamic, interactive dialogue. This task assesses the ability to answer specific questions about one or more prompted regions, *directly measuring its comprehension* rather than its descriptive fluency. To provide a comprehensive and multi-faceted evaluation of the reasoning abilities, we divide it into two distinct but complementary sub-tasks: "*perception*" and "*reasoning*".

**Perception** evaluates the model's foundational ability to recognize basic visual attributes of a single object, serving as a litmus test for its core visual acuity. This task quantifies the ability to perceive the foundational details. Specifically, for a given visual prompt, the model is asked targeted questions about its intrinsic visual properties, specifically focusing on color, shape, material, and texture/pattern.

**Reasoning** is designed to probe higher-order cognitive abilities. This component challenges the model to synthesize information from local prompts, global context, and the relationships between multiple prompts to arrive at logical conclusions. It is composed of several sub-tasks, each targeting a unique and challenging aspect of visual reasoning:

- **Position** evaluates the model's grasp of spatial arrangement and ordinal logic within a global context. A model is presented with a mask on a single object within a larger group and asked to identify its *precise position* in a complex, grid-like structure. Answering correctly requires the model to not only recognize the masked object but also to process the *entire* scene structure.

- **Non-Entity Recognition** is designed to test this specific capability by requiring the model to leverage sufficient *global context*. For instance, the given prompt might highlight a reflection in a mirror, the shadow of a person, a face depicted on a television screen, and so on. The model is then queried to determine if the prompted region corresponds to a physical entity. Success in this task demonstrates that the model is performing sophisticated context-aware reasoning rather than simple pattern matching on the masked pixels alone.

- **Relation** measures the capacity for complex compositional reasoning across multiple prompts. In this challenging setup, the model is presented with several visual prompts and must deduce the intricate spatial or logical relationship between them. A key challenge is *the inclusion of redundant prompts*. To arrive at the correct answer, the model must ignore the potentially distracting information. It requires the model to build a mental "scene graph", which is essential for comprehending complex object assemblies and interactions in cluttered, real-world environments.

For more benchmark details, including the annotation pipeline and statistics, please refer to Appendix B.1 and Appendix B.2, respectively.

Table 1: Comparison on **GAR-Bench-VQA**. ∗ indicates this subtask evaluates the interaction between multiple visual prompts. † means evaluated with the thinking mode. Our **GAR-1B** even outperforms InternVL3-78B. Moreover, **GAR-8B** surpasses private state-of-the-art non-thinking model GPT-4o.

| Method | Overall | Perception (198) | | | | Reasoning (226) | | |
|---|---|---|---|---|---|---|---|---|
| | | Color (69) | Shape (64) | Texture (29) | Material (36) | Position (64) | Non-Entity* (61) | Relation* (101) |
| *Private General MLLMs* | | | | | | | | |
| GPT-4o | 53.5 | 34.8 | 65.3 | 48.3 | 52.8 | 57.8 | 60.2 | 61.4 |
| o3† | 61.3 | 58.0 | 70.3 | 55.2 | 63.9 | 54.7 | 49.2 | 71.3 |
| Gemini-2.5-Pro† | 64.2 | 62.3 | 68.8 | 58.6 | 66.7 | 64.1 | 64.9 | 70.3 |
| *Public General MLLMs* | | | | | | | | |
| Qwen2.5-VL-3B | 34.4 | 29.0 | 25.0 | 34.5 | 30.6 | 43.8 | 26.2 | 44.6 |
| Qwen2.5-VL-7B | 41.7 | 39.1 | 40.6 | 44.8 | 27.8 | 59.4 | 36.1 | 40.6 |
| Qwen2.5-VL-32B | 50.9 | 46.4 | 53.1 | 41.4 | 30.6 | 71.9 | 36.1 | 58.4 |
| Qwen2.5-VL-72B | 52.8 | 46.4 | 50.0 | 65.5 | 33.3 | 68.8 | 44.3 | 57.4 |
| InternVL3-2B | 35.1 | 30.4 | 21.9 | 48.3 | 38.9 | 48.4 | 26.2 | 38.6 |
| InternVL3-8B | 38.9 | 36.2 | 37.5 | 58.6 | 41.7 | 51.6 | 27.9 | 33.6 |
| InternVL3-38B | 46.5 | 39.1 | 40.6 | 51.7 | 55.6 | 60.9 | 36.1 | 47.5 |
| InternVL3-78B | 50.5 | 44.9 | 54.7 | 58.6 | 61.1 | 53.1 | 47.5 | 45.5 |
| *Region MLLMs* | | | | | | | | |
| Sa2VA-8B | 34.3 | 39.1 | 45.3 | 29.6 | 30.6 | 54.7 | 21.3 | 21.8 |
| VP-SPHINX-13B | 37.5 | 33.3 | 25.0 | 44.8 | 38.9 | **60.9** | 34.3 | 32.7 |
| DAM-3B | 38.2 | 55.1 | 39.1 | 41.4 | 36.1 | 31.3 | 36.1 | 31.7 |
| PAM-3B‡ | 2.4 | 2.9 | 3.1 | 6.9 | 5.6 | 1.6 | 1.6 | 0.0 |
| **GAR-1B** | 50.6 | 55.1 | 46.9 | 69.0 | 47.2 | 21.9 | **62.3** | 56.4 |
| **GAR-8B** | **59.9** | **59.4** | **54.7** | **75.9** | **52.8** | 48.4 | 60.7 | **68.3** |

Table 2: Comparison of **localized relational captioning** on our **GAR-Bench-Cap**. We utilize GPT-4o (OpenAI, 2024a) with cropped images and masks to judge the correctness of the answer.

| Method | Overall (204) | Simple (97) | Detailed (107) |
|---|---|---|---|
| *Private General MLLMs* | | | |
| GPT-4o | 51.5 | 39.2 | 62.6 |
| o3 | 56.9 | 37.1 | 74.8 |
| Gemini-2.5-Pro | 59.3 | 51.6 | 66.4 |
| *Public General MLLMs* | | | |
| Qwen2.5-VL-3B | 22.5 | 9.3 | 34.6 |
| Qwen2.5-VL-7B | 32.4 | 12.4 | 50.5 |
| Qwen2.5-VL-32B | 36.8 | 17.5 | 54.3 |
| InternVL3-2B | 29.4 | 14.4 | 43.0 |
| InternVL3-8B | 33.8 | 11.3 | 54.2 |
| InternVL3-38B | 45.1 | 29.9 | 58.9 |
| *Region MLLMs* | | | |
| DAM-3B | 13.1 | 17.5 | 10.3 |
| PAM-3B | 21.1 | 3.1 | 39.3 |
| VP-SPHINX-13B | 32.3 | 27.8 | 39.3 |
| Sa2VA-8B | 45.6 | 46.4 | 44.9 |
| **GAR-1B** | 57.5 | 56.7 | 63.6 |
| **GAR-8B** | **62.2** | **66.0** | **64.5** |

Table 3: Comparison on **detailed localized captioning** on DLC-Bench (Lian et al., 2025). † indicates using GPT-4o (OpenAI, 2024a) with extra cropped images as judge, otherwise performing *text-only* judging, where discussions can be found in Appendix F. ‡ means our evaluation with the official checkpoint.

| Method | Avg. | Pos. | Neg. |
|---|---|---|---|
| *Private General MLLMs* | | | |
| Gemini-2.5-Pro | 55.8 | 36.5 | 75.2 |
| GPT-4o | 61.5 | 43.4 | 79.6 |
| o1 | 62.5 | 46.3 | 78.8 |
| *Region MLLMs* | | | |
| Shikra-7B | 22.2 | 2.7 | 41.8 |
| Ferret-7B | 22.4 | 6.4 | 38.4 |
| RegionGPT-7B | 27.2 | 13.0 | 41.4 |
| VP-SPHINX-13B | 22.5 | 11.7 | 33.2 |
| DAM-3B | 64.5‡ | 47.2‡ | 81.8‡ |
| **GAR-1B** | **67.9** | 48.9 | **87.0** |
| **GAR-8B** | 67.4 | **50.2** | 84.6 |
| DAM-3B† | 72.6‡ | 61.8‡ | 83.4‡ |
| **GAR-1B†** | **77.1** | 66.2 | **88.0** |
| **GAR-8B†** | 77.0 | **68.0** | 86.0 |

## 4 EXPERIMENTS

Owing to page limitations, we only present the key properties in this section. For implementation details, comparative baselines, and ablation studies, please refer to Appendix C.

Table 4: *Zero-shot* results on region-level **detailed image captioning** on Ferret-Bench (You et al., 2023) and and MDVP-Bench (Lin et al., 2025a). We adopt SAM (Kirillov et al., 2023) to produce masks conditioned on bounding boxes for MDVP-Bench (Lin et al., 2025a). All results are our reproduction using the official checkpoint, as the original judger GPT-4V is no longer available, and we take GPT-4o as the judge.

Table 5: Results of **category-level image recognition** on LVIS (Gupta et al., 2019) and PACO (Ramanathan et al., 2023).

| Method | Ferret-Bench | | MDVP-Bench (Box Caption) | | | |
|---|---|---|---|---|---|---|
| | Refer. | Desc. | Natural | OCR | Multi-Panel | Screenshot |
| Osprey-7B | – | | 107.7 | 99.4 | 70.0 | 81.3 |
| PAM-3B | 52.2 | | 71.4 | 94.3 | 86.8 | 84.5 |
| DAM-3B | 55.0 | | 87.0 | 127.7 | 79.4 | 76.4 |
| **GAR-1B** | 56.0 | | 152.6 | **149.6** | 103.7 | 115.3 |
| **GAR-8B** | **64.8** | | **178.6** | 149.1 | **117.2** | **123.0** |

| Method | LVIS | | PACO | |
|---|---|---|---|---|
| | Sim. | IoU | Sim. | IoU |
| Shikra-7B | 49.7 | 19.8 | 43.6 | 11.4 |
| GPT4RoI-7B | 51.3 | 12.0 | 48.0 | 12.1 |
| Ferret-7B | 63.8 | 36.6 | 58.7 | 26.0 |
| Osprey-7B | 65.2 | 38.2 | 73.1 | 52.7 |
| DAM-8B | 89.0 | 77.7 | 84.2 | 73.2 |
| PAM-3B | 88.6 | 78.3 | 87.4 | 74.9 |
| **GAR-1B** | 91.0 | 68.2 | 93.2 | 72.4 |
| **GAR-8B** | **93.6** | **88.7** | **95.5** | **91.8** |

Table 6: *Zero-shot* comparison of **detailed localized video captioning** on VideoRefer-Bench$^D$ (Yuan et al., 2025b). For "single-frame", we select the target frame and apply AnyRes with `max_num_tiles=16`. For "multi-frame", we uniformly sample 16 frames and turn off AnyRes.

| Method | Single-Frame | | | | | Multi-Frame | | | | |
|---|---|---|---|---|---|---|---|---|---|---|
| | Avg. | SC | AD | TD | HD | Avg. | SC | AD | TD | HD |
| *General MLLMs* | | | | | | | | | | |
| LLaVA-OneVison-7B | 2.12 | 2.62 | 1.58 | 2.19 | 2.07 | 2.48 | 3.09 | 1.94 | 2.50 | 2.41 |
| Qwen2-VL-7B | 2.39 | 2.97 | 2.24 | 2.03 | 2.31 | 2.55 | 3.30 | 2.54 | 2.22 | 2.12 |
| InternVL2-26B | 2.84 | 3.55 | 2.99 | 2.57 | 2.25 | 3.20 | 4.08 | 3.35 | 3.08 | 2.28 |
| GPT-4o | 2.95 | 3.34 | 2.96 | 3.01 | 2.50 | 3.25 | 4.15 | 3.31 | 3.11 | 2.43 |
| *Region MLLMs* | | | | | | | | | | |
| Elysium-7B | 1.57 | 2.35 | 0.30 | 0.02 | **3.59** | – | – | – | – | – |
| Ferret-7B | 2.18 | 3.08 | 2.01 | 1.54 | 2.14 | 2.23 | 3.20 | 2.38 | 1.97 | 1.38 |
| Osprey-7B | 2.34 | 3.19 | 2.16 | 1.54 | 2.45 | 2.41 | 3.30 | 2.66 | 2.10 | 1.58 |
| Artemis-7B | – | – | – | – | – | 2.26 | 3.42 | 1.34 | 1.39 | 2.90 |
| DAM-8B | – | – | – | – | – | 3.34 | 4.45 | **3.30** | **3.03** | 2.58 |
| **GAR-1B** | 2.72 | **4.41** | **2.98** | 1.09 | 2.40 | 2.83 | 4.38 | 3.01 | 1.61 | 2.30 |
| **GAR-8B** | **2.75** | **4.41** | 2.96 | 1.58 | 2.45 | **3.44** | **4.53** | 3.25 | 2.57 | **3.42** |

**Advanced comprehension** requires precisely modeling complex relationships between multiple prompts. To evaluate this capability, we conducted a comprehensive comparison on our GAR-Bench-VQA. As demonstrated in Table 1, GAR-8B achieves an impressive overall score of 54.5, surpassing even the powerful, private, state-of-the-art non-thinking model, GPT-4o (OpenAI, 2024a). Furthermore, the efficiency and effectiveness of our approach are highlighted by GAR-1B. Despite its significantly smaller size, it scores 50.6 overall, outperforming large-scale public models like InternVL3-78B (Zhu et al., 2025). This advantage is particularly evident in fine-grained perception tasks, where GAR-1B and GAR-8B achieve "Texture" scores of 69.0 and 75.9, respectively.

**Detailed localized captioning** requires generating detailed descriptions for given regions with multiple sentences. We benchmark our GAR models on a series of challenging datasets, and the results consistently demonstrate their state-of-the-art capabilities. As shown in Table 2, on our GAR-Bench-Cap, GAR-1B and GAR-8B achieve the highest overall scores of 57.5 and 62.2, respectively, even exceeding that of powerful private models like Gemini-2.5-Pro (DeepMind, 2025b). This superiority is further confirmed on the DLC-Bench (Lian et al., 2025) in Table 3, where GAR-1B and GAR-8B again outperform top models like DAM-3B using either LLaMA3.1 (Grattafiori et al., 2024) or GPT-4o (OpenAI, 2024a) as the judge. The zero-shot performance of our models on Ferret-Bench (You et al., 2023) and MDVP-Bench (Lin et al., 2025a), detailed in Table 4, is particularly noteworthy. On both benchmarks, our GAR emerges as the top-performing model across every single category. Specifically on MDVP-Bench, our models show a commanding lead, with GAR-8B achieving a score of 178.6 on natural images, a result that is substantially higher than any competitor. Collectively, these comprehensive evaluations across multiple benchmarks unequivocally establish **GAR** as the new state-of-the-art for producing rich, accurate, and detailed localized captions.

Table 7: *Zero-shot* comparison of **detailed video understanding** on VideoRefer-Bench[Q] (Yuan et al., 2025b). † indicates trained on *in-domain* VideoRefer-700k with regard to VideoRefer-Bench. Notably, *our zero-shot GAR-8B even outperforms in-domain VideoRefer-7B.*

| Method | Overall (1000) | Basic Questions (235) | Sequential Questions (256) | Relationship Questions (252) | Reasoning Questions (143) | Future Predictions (114) |
|---|---|---|---|---|---|---|
| *General MLLMs* | | | | | | |
| InternVL2-26B | 65.0 | 58.5 | 63.5 | 53.4 | 88.0 | 78.9 |
| Qwen2-VL-7B | 66.0 | 62.0 | 69.6 | 54.9 | 87.3 | 74.6 |
| LLaVA-OneVision-7B | 67.4 | 58.7 | 62.9 | 64.7 | 87.4 | 76.3 |
| GPT-4o | 71.3 | 62.3 | 74.5 | 66.0 | 88.0 | 73.7 |
| *Region MLLMs* | | | | | | |
| Osprey-7B | 39.9 | 45.9 | 47.1 | 30.0 | 48.6 | 23.7 |
| Ferret-7B | 48.8 | 35.2 | 44.7 | 41.9 | 70.4 | 74.6 |
| VideoRefer-7B† | 71.9 | 75.4 | 68.6 | 59.3 | **89.4** | **78.1** |
| **GAR-1B** | 69.9 | 75.0 | 69.9 | 59.7 | 83.2 | 63.7 |
| **GAR-8B** | **72.0** | **77.2** | **71.0** | **61.7** | 86.6 | 68.1 |

**Open-class category-level image recognition** requires the model to recognize the category of the object and part entities. We evaluate this capability in Table 5. Our GAR-8B demonstrates a significant leap in performance, establishing a new state-of-the-art. It consistently outperforms all prior methods across every metric, achieving top scores of 93.6 semantic similarity and 88.7 semantic IoU on LVIS (Gupta et al., 2019), and 95.5 semantic similarity and 91.8 semantic IoU on PACO (Ramanathan et al., 2023). This indicates its superior ability in both semantic understanding and precise localization. These results demonstrate the effectiveness of GAR for complex recognition tasks, showcasing its robust performance in identifying a diverse range of object categories.

**Extension to videos** is straightforward. Similar to Lian et al. (2025), we simply extend our GAR models to videos and evaluate them on VideoRefer-Bench[D] (Yuan et al., 2025b) and VideoRefer-Bench[Q] (Yuan et al., 2025b) in Table 6 and Table 7, respectively. We uniformly sample 16 frames to represent a video. Our GAR-8B surpasses DAM-8B (Lian et al., 2025) under the zero-shot setting. More importantly, as demonstrated in Table 7, our *our zero-shot GAR-8B even outperforms in-domain VideoRefer-7B*, demonstrating its strong comprehension capabilities can be easily transferred to videos. However, as our models are actually trained with images, they get reasonably low scores on temporally related tasks, *e.g.*, temporal description (TD) in Table 6 and future predictions in Table 7.

**Qualitative Results.** We provide qualitative comparisons between our **GAR-8B** with DAM-3B (Lian et al., 2025) on detailed localized captioning on DLC-Bench (Lian et al., 2025) in Figure 5. As demonstrated in the figure, our **GAR-8B** is more capable of generating precise descriptions, especially when the category of the given prompt can be determined only when understanding sufficient global contexts. More comparisons can be found in Appendix D.

## 5 CONCLUSION

This paper introduces **G**rasp **A**ny **R**egion (**GAR**), a family of MLLMs for region understanding, and **GAR-Bench**, a systematic evaluation framework that not only provides a more accurate evaluation of single-region comprehension, but also for multi-prompt interaction and advanced compositional reasoning. On detailed captioning benchmarks (Lian et al., 2025; You et al., 2023; Lin et al., 2025a), **GAR** demonstrates superior performance over DAM (Lian et al., 2025). More importantly, our **GAR** achieves advanced comprehension capability in modeling interactions between multiple prompts. Specifically, on GAR-Bench-VQA, **GAR-1B** even surpasses InternVL3-78B (Zhu et al., 2025). On VideoRefer-Bench[Q] (Yuan et al., 2025b), our *zero-shot* **GAR-8B** even outperforms in-domain VideoRefer-7B (Yuan et al., 2025b). We hope our work inspires the community to develop MLLMs that can perceive, interrogate, and understand the dense visual world more effectively.

## ACKNOWLEDGEMENTS

This work was supported by the Beijing Natural Science Foundation (No. L257015) and the National Natural Science Foundation of China (No. 62320106010).

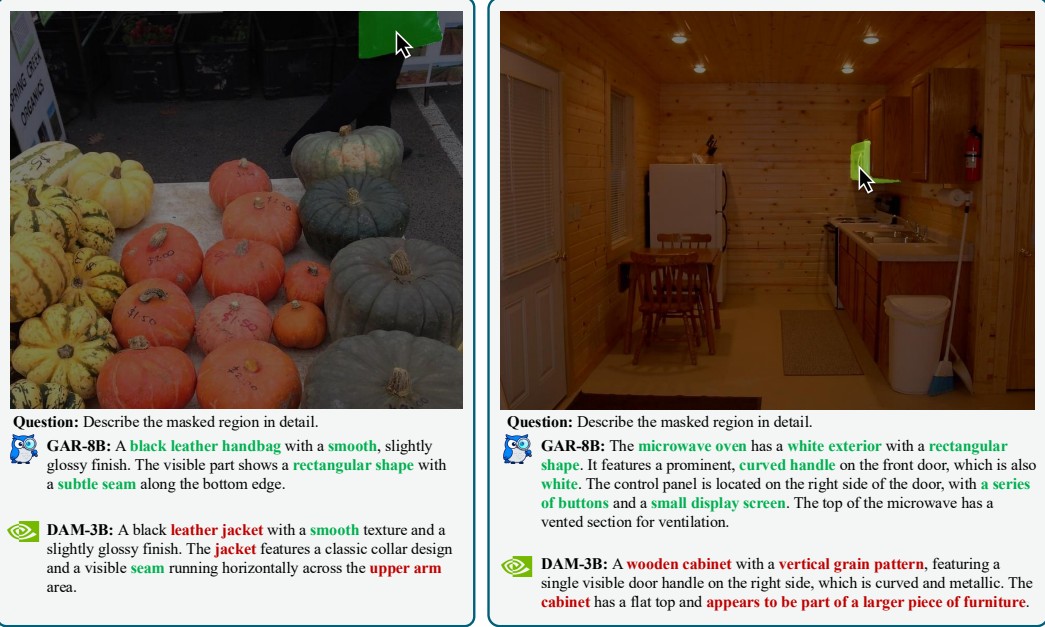

Figure 5: **Qualitative comparisons** on DLC-Bench (Lian et al., 2025), where green indicates correct descriptions and red means errors.

## ETHICS STATEMENT

Our research is grounded in ethical practices, with particular attention paid to the responsible use of data. All datasets employed in this study are publicly available and well-established within the computer vision community. Specifically, our training data includes ImageNet-21K (Deng et al., 2009) and the PSG (Yang et al., 2022) dataset (with image sources from COCO (Lin et al., 2014), while our benchmarking was conducted on FSC-147 (Ranjan et al., 2021), RGBD-Mirror (Mei et al., 2021), and SA-1B (Kirillov et al., 2023). Our use of this data is in accordance with their provided licenses and intended academic purpose.

## REPRODUCIBILITY STATEMENT

We are committed to ensuring the reproducibility of the research presented in this paper. To this end, comprehensive implementation details for our models and experiments are provided in Appendix C, including the training procedures and all hyperparameters used. Furthermore, upon acceptance of this paper, all source code, datasets, and trained model checkpoints will be made publicly available.

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

APPENDIX

## A  OVERVIEW

Here is the table of contents of this appendix:

- In Appendix B, we introduce details of our **GAR-Bench**, including the annotation pipeline and statistics.

- In Appendix C, we provide more implementation details as well as experimental results. Detailed ablations of each component can be found in this section.

- In Appendix D, we provide qualitative results on both detailed *image* captioning and understanding, and localized *video* captioning and understanding.

- In Appendix E, we discuss potential limitations and analyze failure cases.

- In Appendix F, we discuss some underlying issues towards the evaluation protocols of DLC-Bench (Lian et al., 2025).

- In Appendix G, we provide all prompts we utilized to construct our dataset.

- Finally in Appendix H, we discuss the use of LLMs in preparing this paper.

## B  DETAILS OF GAR-BENCH

### B.1  ANNOTATION PIPELINE

The construction of **GAR-Bench** follows a rigorous, semi-automated pipeline designed to generate high-quality, diverse, and challenging data. This process combines the strengths of advanced foundation models for initial data generation with the nuanced judgment of a team of 8 MLLM experts for curation, annotation, and quality control.

**Image Selection.** To ensure the relevance and challenge of our sub-tasks, we begin by carefully curating source images from existing datasets known to contain specific visual patterns. For the "*relation*" tasks, we source images from the Panoptic Scene Graph (PSG) dataset (Yang et al., 2022), which is rich in complex scene graphs and explicit object relationships, providing a natural foundation for multi-prompt interaction queries. For the "*non-entity recognition*" task, we utilize the RGBD-Mirror dataset (Mei et al., 2021), as it specifically contains scenes with mirrors and reflections, allowing us to create unambiguous test cases for distinguishing real objects from illusory ones. For the "position" task, we select images from the FSC-147 dataset (Ranjan et al., 2021), which features images with numerous countable objects often arranged in grid-like patterns, making it ideal for evaluating spatial and ordinal reasoning. Other images are from SA-1B (Kirillov et al., 2023).

**Mask Labeling.** Following image selection, we generate high-quality segmentation masks for all potential objects of interest. This stage is similar to Li et al. (2025), which decomposes complex scenes into different objects, while not containing numerous meaningless, trivial objects like those in the SA-1B (Kirillov et al., 2023) dataset.

**Object Selection and Annotation.** With a high-quality pool of object masks generated, the annotation team performs the critical tasks of selection and annotation. The experts first reviewed the masks, selecting only those with high segmentation quality that are also qualified for the target sub-task. Concurrently, they are responsible for annotating the ground-truth information required for the benchmark. Specifically, for the "reasoning" protocol of **GAR-Bench-VQA**, they meticulously annotate the correct answers for relation, ordering, and entity status. For **GAR-Bench-Cap**, they annotate the ground-truth captions describing the interactions between the selected masked objects.

**Automated Attribute Generation.** For the "perception" protocol of **GAR-Bench-VQA**, we leverage the advanced capabilities of Gemini-2.5-Pro (DeepMind, 2025b). For each selected and verified object mask, we prompt the model to generate a list of its basic perceptual attributes, including its primary color, shape, material, and any discernible texture or pattern.

**Quality Control and Formatting.** The raw, annotated data then underwent a meticulous, multi-stage quality control process. First, human experts review all machine-generated attributes from the previous step to verify their factual correctness and filter out any ambiguous or inaccurate labels. Following this verification, the experts transform the raw annotations into the final benchmark formats.

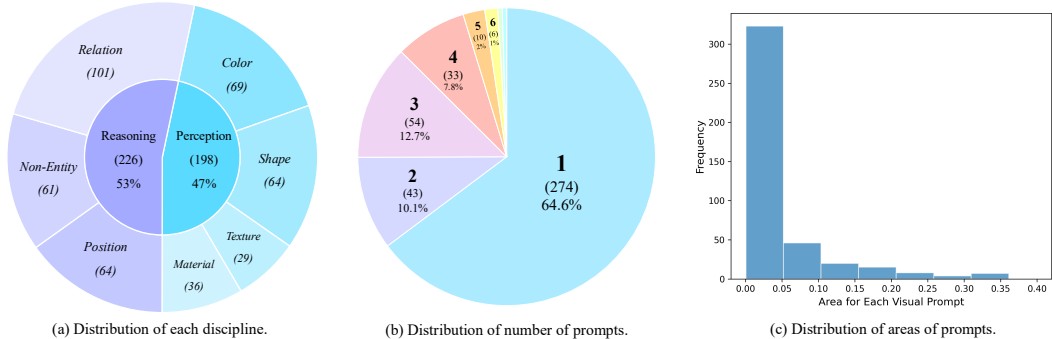

(a) Distribution of each discipline.    (b) Distribution of number of prompts.    (c) Distribution of areas of prompts.

Figure 6: **Statistics of our GAR-Bench**. We (a) slightly prioritize reasoning over perception, and build challenging questions through (b) multiple visual prompts (even have 2 questions with 7 prompts and 9 prompts) and (c) small areas of each prompt with an average of 4.4%.

For all VQA tasks, they rewrite the question-answer pairs into a standardized multiple-choice format, ensuring consistent and objective evaluation. For the captioning task, the ground-truth data was structured for compatibility with LLM-as-a-Judge evaluation protocols similar to Lian et al. (2025).

**Difficulty Filtering.** As a final quality assurance measure, we implement a difficulty filtering process to ensure the benchmark remains challenging for even the most advanced models. Specifically, any question answered correctly by *all* four state-of-the-art non-thinking MLLMs, *i.e.*, Qwen2.5-VL-72B (Bai et al., 2025), InternVL3-78B (Zhu et al., 2025), GPT-4o (OpenAI, 2024a), and Gemini-2.5-Flash (DeepMind, 2025a), was excluded from the final benchmark.

### B.2 STATISTICS

**Distribution of Each Discipline.** As demonstrated in Figure 6a, **GAR-Bench** slightly prioritizes advanced reasoning (53%) over basic perception (47%) with a relatively balanced distribution. In addition, it prioritizes complex relational reasoning with *multiple prompts* in the "*relation*" protocol.

**Distribution of Number of Prompts.** As illustrated in Figure 6b, our **GAR-Bench** even contains 2 questions with 7 prompts and 9 prompts, respectively, leading to an advanced requirement of modeling complex relationships between multiple visual prompts.

**Distribution of Areas of Prompts.** We compute the *relative* area of each visual prompt in Figure 6c, where the majority of prompts in **GAR-Bench** are extremely small, with a sharp peak near 0.0. The mean area across all questions is 4.4%. This distribution highlights the importance of addressing small-scale and fine-grained understanding.

## C   MORE EXPERIMENTS

**Implementation Details.** We adopt PerceptionLM series (Cho et al., 2025) as our base model, as it demonstrates strong perception capabilities among several open-source MLLMs. We perform supervised fine-tuning of the model on our GAR-2.5M using Xtuner (Contributors, 2023) with the AdamW optimizer (Loshchilov & Hutter, 2017) with a global batch size of 64 and a learning rate of 1e-5 with a cosine decay (Loshchilov & Hutter, 2016).

**Comparison Baselines.** We mainly compare our **GAR** with both general MLLMs, including state-of-the-art private models (OpenAI, 2024a; 2025; DeepMind, 2025b), and representative public models (Bai et al., 2025; Zhu et al., 2025; Liu et al., 2023), and region-level MLLMs, including GLaMM (Rasheed et al., 2024), GPT4RoI (Zhang et al., 2024a), Osprey (Yuan et al., 2024), Shikra (Chen et al., 2023), Ferret (You et al., 2023), RegionGPT (Guo et al., 2024), OMG-LLaVA (Zhang et al., 2024b), VP-SPHINX (Lin et al., 2025a), Sa2VA (Yuan et al., 2025a), DAM (Lian et al., 2025), and PAM (Lin et al., 2025b). We transform masks to boxes for box-level MLLMs, *e.g.*, (Zhang et al., 2024a; Chen et al., 2023; You et al., 2023; Lin et al., 2025b), as our **GAR-Bench** provides segmentation masks by default. On video benchmarks, we further compare with LLaVA-OneVision (Li et al., 2024), Qwen2-VL (Wang et al., 2024b), InternVL2 (Chen et al., 2024c), Elysium (Wang et al., 2024a), Artemis (Qiu et al., 2024), and VideoRefer (Yuan et al., 2025b).

Table 8: **Ablations across different model architectures** with PerceptionLM-1B. † indicates using GPT-4o (OpenAI, 2024a) with extra cropped images as the judge, instead of text-only judging. Our proposed RoI-aligned feature replay strategy effectively preserves necessary global contexts. We also report the average latency (ms) to generate the first token and the maximum number of tokens for ViT (Dosovitskiy et al., 2021). By default, we set `max_num_tiles=16` for AnyRes (Liu et al., 2024), resulting in a maximum of 17 crops in total for one global image.

| | Global | Local | GAR-Bench | | DLC-Bench† | | | Inference Speed | |
|---|---|---|---|---|---|---|---|---|---|
| | | | Caption | VQA | Avg. | Pos. | Neg. | Latency | # ViT Tokens |
| ① | – | image + mask | 20.1 | 37.8 | 69.3 | 60.2 | 78.4 | 36.1 | 256 |
| ② | – | image + mask + cross-attention | 19.1 | 40.0 | 68.8 | 57.3 | 80.3 | 57.1 | 4,608 |
| ③ | image + mask | image + mask | 28.4 | 36.6 | **77.4** | **70.1** | 84.8 | 93.1 | 4,608 |
| ④ | image + mask | RoI-aligned feature replay | **57.5** | **50.6** | 77.1 | 66.2 | **88.0** | 87.7 | 4,352 |
| ⑤ | image | RoI-pooled feature | 39.3 | 39.1 | 50.0 | 37.3 | 62.7 | 78.2 | 4,352 |
| ⑥ | image | RoI-aligned feature replay | 42.1 | 40.1 | 67.1 | 55.8 | 78.4 | 90.8 | 4,608 |
| ⑦ | – | image | 17.8 | 35.4 | 65.0 | 59.3 | 70.7 | 30.4 | 256 |
| ⑧ | image + mask | – | 32.7 | 40.4 | 51.7 | 40.4 | 63.0 | 75.0 | 4,352 |

Table 9: **Ablations across different model architectures** with different base models. † indicates using GPT-4o (OpenAI, 2024a) with extra cropped images as the judge, instead of text-only judging. Our proposed RoI-aligned feature replay strategy effectively preserves necessary global contexts.

| | Global | Local | GAR-Bench | | DLC-Bench† | | |
|---|---|---|---|---|---|---|---|
| | | | Caption | VQA | Avg. | Pos. | Neg. |
| *Base Model: Qwen2.5-VL-3B* | | | | | | | |
| ⑤ | – | image + mask | 24.5 | 30.7 | 52.2 | 38.0 | 66.4 |
| ⑥ | – | image + mask + cross-attention | 27.9 | 30.0 | 55.7 | 46.8 | 64.6 |
| ⑦ | image + mask | image + mask | 34.3 | 32.1 | 62.1 | 50.7 | 73.5 |
| ⑧ | image + mask | RoI-aligned feature replay | 41.2 | 40.8 | 69.2 | 58.1 | 80.3 |
| *Base Model: InternVL3-2B* | | | | | | | |
| ⑨ | – | image + mask | 24.6 | 33.0 | 65.6 | 48.5 | 82.6 |
| ⑩ | – | image + mask + cross-attention | 29.4 | 31.8 | 68.8 | 56.7 | 80.9 |
| ⑪ | image + mask | image + mask | 32.8 | 36.1 | 70.3 | 61.6 | 79.0 |
| ⑫ | image + mask | RoI-aligned feature replay | 43.1 | 44.6 | 73.0 | 63.8 | 82.2 |

**Ablations on Architecture Designs.** We first elaborate on our key architecture design, *i.e.*, RoI-aligned feature replay in Table 8. Other baselines include: ① only local images, ② DAM-like architectures (Lian et al., 2025) which preserves context via zero-initialized gated cross-attention, ③ simply cropping local images as a supplement of global images, and ④ our RoI-aligned feature replay design. As demonstrated in Table 8, both ①, ②, and ③ struggle at modeling multi-prompt relations, leading to poor results on GAR-Bench, although ③ is superior at precise description on DLC-Bench (Lian et al., 2025). However, our proposed RoI-aligned feature replay strategy effectively preserves necessary global contexts while achieving competitive performances on DLC-Bench.

In Table 9, we further extend our ablations on model architectures to more base models, including Qwen2.5-VL-3B (Bai et al., 2025) and InternVL3-2B (Zhu et al., 2025). As demonstrated in the table, our proposed RoI-aligned feature replay *consistently* brings significant improvements over different base models.

**Ablations on Data Pipeline.** We study the effectiveness of our data in Table 10. Starting from the seed dataset, *i.e.*, Describe-Anything-1.5M (Lian et al., 2025), we first add our Fine-Grained Dataset-456K, and then add our Relation Dataset-414K. By introducing our Fine-Grained Dataset-456K, our model is able to produce more accurate recognition, leading to an improvement of +3.1 on DLC-Bench (Lian et al., 2025). By further combining our proposed Relation Dataset-414K, the model is finally equipped with compositional reasoning capabilities with multiple prompts at this time, resulting in significant improvements on our GAR-Bench.

**Performances on General Multimodal Benchmarks.** We compare our **GAR-8B** with other region-level models, *i.e.*, DAM-3B (Lian et al., 2025) and PAM-3B (Lin et al., 2025b), on general vision-centric multimodal benchmarks, including V* (Wu & Xie, 2024), MMVP (Tong et al., 2024b),

Table 10: **Ablations on each component of our data** with 1B model size. † indicates using GPT-4o (OpenAI, 2024a) with extra cropped images as the judge, instead of text-only judging. Each component of our data plays a significant role.

| Data | | GAR-Bench | | DLC-Bench† | | |
|---|---|---|---|---|---|---|
| | | Caption | VQA | Avg. | Pos. | Neg. |
| ① | Seed Dataset-1.5M | 13.8 | 41.5 | 74.4 | 63.0 | 85.8 |
| ② | ① + Fine-Grained Dataset-456K | 14.2 | 44.1 | **77.5** | **67.6** | 87.4 |
| ③ | ② + Relation Dataset-414K | **57.5** | **50.6** | 77.1 | 66.2 | **88.0** |

Table 11: Performance on general multimodal benchmarks (Wu & Xie, 2024; Tong et al., 2024b; xAI, 2024; Chen et al., 2024a), where we set mask = 1 for evaluation. Our **GAR** maintains the most general performance. Combining general VQA datasets would be effective.

| Method | V* | MMVP | RealWorldQA | MMStar |
|---|---|---|---|---|
| DAM-3B (Lian et al., 2025) | 45.0 | 60.7 | 54.3 | 39.7 |
| PAM-3B (Lin et al., 2025b) | 1.4 | 4.3 | 1.7 | 2.7 |
| *Base Model: PerceptionLM-8B* | | | | |
| PercetionLM-8B (Cho et al., 2025) | 69.1 | 76.0 | 75.0 | 57.1 |
| **GAR-8B** | 59.2 | 78.0 | 58.7 | 43.9 |
| **GAR-8B** (w/ 600K General Data (Li et al., 2024)) | 62.3 | 79.7 | 61.8 | 51.6 |

Table 12: **Robustness analysis of GAR-Bench-VQA**, where we randomly sample a subset of each subtask. *The relative ordering is stable.*

| Model | Full (424) | | 1/2 (212) | | 1/4 (106) | |
|---|---|---|---|---|---|---|
| | Overall | Rank | Overall | Rank | Overall | Rank |
| PAM-3B | 2.4 | 9 | 4.3 | 9 | 0.9 | 9 |
| VP-SPHINX-13B | 37.5 | 8 | 40.0 | 8 | 33.3 | 8 |
| DAM-3B | 38.2 | 7 | 48.6 | 7 | 41.9 | 7 |
| GAR-1B | 50.6 | 6 | 51.4 | 6 | 49.5 | 5 |
| Qwen2.5-VL-32B | 50.9 | 5 | 52.4 | 5 | 48.6 | 6 |
| GPT-4o | 53.5 | 4 | 56.7 | 4 | 57.1 | 4 |
| GAR-8B | 59.9 | 3 | 60.0 | 3 | 63.8 | 1 |
| o3 | 61.3 | 2 | 63.3 | 1 | 58.1 | 3 |
| Gemini-2.5-Pro | 64.2 | 1 | 61.0 | 2 | 60.0 | 2 |

Table 13: **Robustness analysis of GAR-Bench-Cap**, where we randomly sample a subset of *each subtask*. *The relative ordering remains stable.*

| Model | Full (214) | | 1/2 (107) | | 1/4 (53) | |
|---|---|---|---|---|---|---|
| | Overall | Rank | Overall | Rank | Overall | Rank |
| DAM-3B | 13.1 | 9 | 13.8 | 9 | 14.0 | 9 |
| PAM-3B | 21.1 | 8 | 18.8 | 8 | 20.1 | 8 |
| VP-SPHINX-13B | 32.3 | 7 | 29.7 | 7 | 20.0 | 7 |
| Qwen2.5-VL-32B | 36.8 | 6 | 32.7 | 6 | 26.1 | 6 |
| GPT-4o | 51.5 | 5 | 45.5 | 5 | 52.1 | 4 |
| o3 | 56.9 | 4 | 50.6 | 4 | 50.3 | 5 |
| GAR-1B | 57.5 | 3 | 51.5 | 3 | 53.9 | 3 |
| Gemini-2.5-Pro | 59.3 | 2 | 54.4 | 2 | 58.1 | 2 |
| GAR-8B | 62.2 | 1 | 57.4 | 1 | 61.8 | 1 |

RealWorldQA (xAI, 2024), and MMStar (Chen et al., 2024a). As illustrated in Table 11, our GAR-8B outperforms them by a large margin.

**Robust Analysis of GAR-Bench.** We conducted a subsampling stability analysis. We randomly subsampled GAR-Bench-VQA and GAR-Bench-Cap to 50% and 25% of their original sizes (for each subtask) and re-evaluated the full suite of models in Table 12 and Table 13, respectively. Our goal was to test whether the relative performance rankings remained consistent even with significantly fewer samples. The results, presented in the tables, demonstrate a high degree of ranking stability. As the results show, the relative ordering of models is remarkably stable. For example, in GAR-Bench-Cap, the top 3 models (GAR-8B, Gemini-2.5-Pro (DeepMind, 2025b), GAR-1B) and bottom 3 models (VP-SPHINX (Lin et al., 2025a), PAM (Lin et al., 2025b), DAM (Lian et al., 2025)) maintain their general ranking group even at a 1/4 sample size. While minor fluctuations exist among the top-tier models in the VQA 1/4 split (e.g., GAR-8B jumping from 3rd to 1st), the overall performance tiers are preserved.

**Robust Analysis of LLM-Judges.** To directly investigate the consistency and potential bias of the LLM judge, we conducted a new cross-judge validation experiment. We re-evaluated all models on GAR-Bench-Cap using four different powerful LLMs as judges: GPT-4o (OpenAI, 2024a) (our original judge), o3 (OpenAI, 2025), Gemini-2.5-Flash (DeepMind, 2025a), and Gemini-2.5-Pro (DeepMind, 2025b). The results, presented in Table 14, demonstrate a high degree of consistency in model rankings. While the absolute scores fluctuate across different judges, which reflects their inherent stylistic preferences or scoring strictness, *the relative ranking of models is remarkably stable.* Most importantly, our **GAR-8B** *consistently ranks 1st across all four distinct judges*, and our other

Table 14: **Robustness analysis of the LLM-judges** utlized by GAR-Bench-Cap. *The relative ordering remains stable across different judges.*

| Model | GPT-4o | | o3 | | Gemini-2.5-Flash | | Gemini-2.5-Pro | |
|---|---|---|---|---|---|---|---|---|
| | Score | Rank | Score | Rank | Score | Rank | Score | Rank |
| DAM-3B | 13.1 | 10 | 9.3 | 10 | 9.4 | 10 | 0.3 | 10 |
| PAM-3B | 21.1 | 9 | 21.5 | 9 | 25.3 | 9 | 31.3 | 9 |
| VP-SPHINX-13B | 32.3 | 8 | 27.5 | 8 | 30.4 | 8 | 32.7 | 8 |
| Qwen2.5-VL-32B | 36.8 | 7 | 28.1 | 7 | 37.4 | 7 | 41.8 | 7 |
| InternVL3-38B | 45.1 | 6 | 30.8 | 6 | 41.9 | 6 | 47.4 | 6 |
| GPT-4o | 51.5 | 5 | 31.8 | 5 | 46.3 | 5 | 50.6 | 5 |
| o3 | 56.9 | 4 | 37.8 | 3 | 52.8 | 4 | 60.7 | 3 |
| GAR-1B | 57.5 | 3 | 37.4 | 4 | 55.1 | 3 | 59.8 | 4 |
| Gemini-2.5-Pro | 59.3 | 2 | 44.5 | 2 | 57.4 | 2 | 61.3 | 2 |
| GAR-8B | 62.2 | 1 | 46.7 | 1 | 60.3 | 1 | 62.6 | 1 |

Table 15: **Analysis for general models using four different region-specification formats.** "VQA" and "Cap" represent "GAR-Bench-VQA" and "GAR-Bench-Cap", respectively. Powerful, general-purpose VLMs consistently struggle across *all* four settings.

| Input Type | GPT-4o | | | o3 | | | Gemini-2.5-Pro | | |
|---|---|---|---|---|---|---|---|---|---|
| | VQA | Cap | DLC-Bench | VQA | Cap | DLC-Bench | VQA | Cap | DLC-Bench |
| Type 1 | 53.5 | 51.5 | 41.0 | 61.3 | 56.9 | 48.0 | 64.2 | 59.3 | 48.4 |
| Type 2 | 56.4 | 52.0 | 38.4 | 63.4 | 54.9 | 47.8 | 61.9 | 58.1 | 52.9 |
| Type 3 | 51.2 | 31.9 | 25.4 | 57.8 | 40.7 | 34.9 | 61.9 | 61.3 | 54.6 |
| Type 4 | 48.3 | 38.2 | 28.5 | 59.0 | 47.1 | 41.3 | 66.0 | 44.1 | 47.8 |

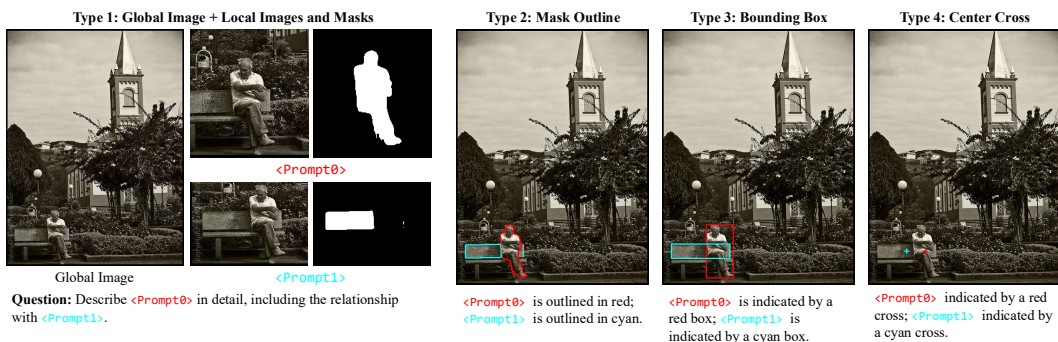

Figure 7: **Illustration of different input types.**

GAR models also consistently place in the top tier. This experiment suggests that our primary claims about GAR's superior performance are robust and not an artifact of a single judge's bias.

**Input Type Analysis for General Models.** To investigate whether general models lack the ability to interpret masks or are genuinely deficient in understanding local details and relationships, we conduct an analysis using four different input types illustrated in Figure 7: (1) Type 1: Separated global image and local masks (our original setting). (2) Type 2: Drawing mask outlines directly onto the image. (3) Type 3: Using bounding boxes derived from masks. (4) Type 4: Using center-point crosses derived from masks. Empirical results are presented in Table 15, where powerful models consistently struggle across all four settings. This crucial finding reveals a fundamental deficiency in fine-grained perception and relational reasoning, regardless of how the regions of interest are indicated.

## D  QUALITATIVE RESULTS

### D.1  QUALITATIVE RESULTS ON GAR-BENCH

**"Relation" of GAR-Bench-VQA.** In Figure 8, we provide qualitative comparisons on the "relation" protocol of our GAR-Bench-VQA, including two failure cases (the last row). As demonstrated in the figure, GAR-8B manages to not only effectively model relationships but also leverage crucial local

details for choosing the best answer. For instance, in the right example of the middle row, the person (`<Prompt0>`) is actually *not* reading the book (`<Prompt1>`), since she is looking at the camera. Our GAR-8B manages to recognize such details and thus select "`<Prompt0>` is *holding* `<Prompt1>`" instead of "reading", while both Gemini-2.5-Pro (DeepMind, 2025b) and o3 (OpenAI, 2025) fail.

However, as illustrated in the last two examples in Figure 8, current models still sometimes struggle to understand complex relationships with *more than two objects*. Constructing such complicated training data and keeping the correctness of relation annotations could be a potential solution.

**"Non-Entity Recognition" of GAR-Bench-VQA.** In Figure 9, we provide qualitative comparisons on the "non-entity recognition" protocol of our GAR-Bench-VQA, including two failure cases (the last row). As demonstrated in the figure, GAR-8B is able to correctly recognize objects in the mirror *without* any depth prior, thanks to its encoded global contexts.

However, as demonstrated in the right case in the last row, current models still struggle to distinguish whether the reflection actually comes from the mirror (`<Prompt2>`) or other reflective surfaces (`<Prompt0>` and `<Prompt1>`).

### D.2 QUALITATIVE RESULTS ON VIDEOREFER-BENCH

**Detailed Localized Video Captioning.** In Figure 10, we provide qualitative results of extending GAR-8B to generate detailed *video* descriptions on VideoRefer-Bench[D] (Yuan et al., 2025b). In most cases, where videos usually remain *static*, GAR-8B manages to generate detailed, specific, and precise descriptions. However, as demonstrated in the last example, GAR-8B fails to capture detailed temporal differences among frames, leading to a low score on "temporal description". This is because our GAR models are actually trained with only images and lack *fine-grained* temporal comprehension capabilities.

**Detailed Video Understanding.** In Figure 11, we provide qualitative results of extending GAR-8B to detailed *video* understanding on VideoRefer-Bench[Q]. GAR-8B is capable of understanding basic motions under a *zero-shot* setting, *e.g.*, the sequential question, the relation question, and the reasoning question. However, on the "future prediction" protocol, GAR-8B sometimes fails to choose the best choice with *significant* motion changes.

## E  LIMITATION AND FAILURE CASES

One potential limitation is that our **GAR** is limited to static images. Although it can be successfully extended to video and even achieves competitive results compared with video models (please refer to Tables 6 and 7 for detailed experimental results), it sometimes fails when input videos contain significant motion changes. Specifically, as demonstrated in the failure cases in Figures 10 and 11, GAR-8B is superior at comprehending and describing *static* videos, and is also capable of understanding *basic motions*. However, with significant motion changes, GAR-8B sometimes fails. Carefully collecting video training data could be a potential solution.

## F  DISCUSSION ON DLC-BENCH

Our analysis in Figure 12 reveals a significant weakness in the original judger of DLC-Bench (Lian et al., 2025), which relies on a *text-only* LLM, *i.e.*, LLaMA3.1-8B (Grattafiori et al., 2024), for automated scoring. Specifically, a fundamental flaw in the original DLC-Bench (Lian et al., 2025) evaluation lies in its assumption that *semantic categories can be accurately adjudicated within the abstract confines of language space alone*. However, Figure 12 demonstrates that this *text-only* approach is inherently *unreliable* due to the ambiguity of linguistic labels *without* visual contexts. We argue that the image is the only ground truth capable of resolving this ambiguity. Therefore, we provide the image for valid evaluation. To truly assess a model's descriptive power, the judge *must* be multimodal, capable of grounding the generated caption in the visual reality it purports to describe.

## G  PROMPT TEMPLATES

We provide all of our prompts utilized in building our data in Figures 13, 14, 15, and 16.

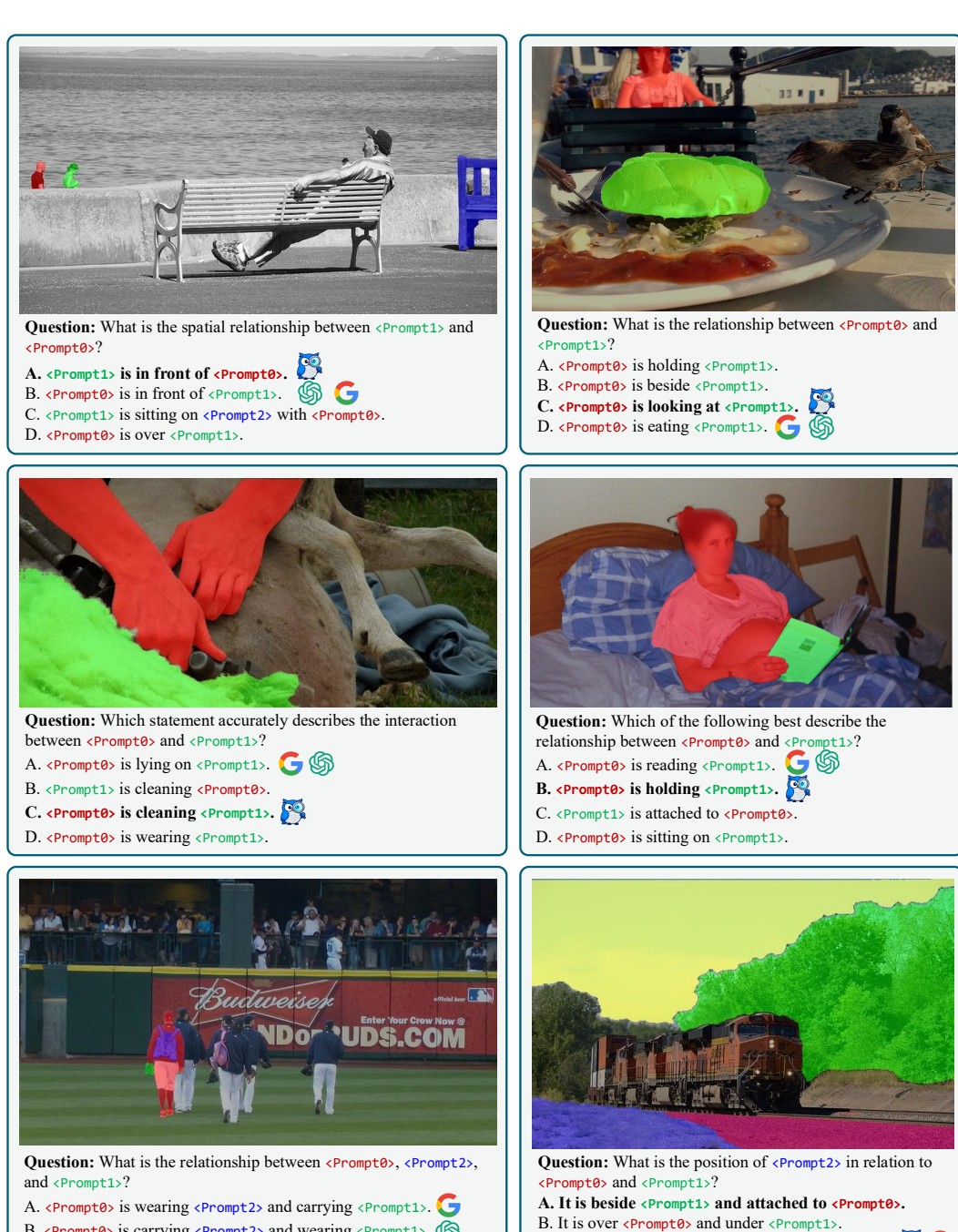

Figure 8: **Qualitative comparisons** on the "relation" protocol of our **GAR-Bench-VQA**, including two failure cases (bottom). Notably, in the right case of the middle row, the person (`<Prompt0>`) is actually *not* reading the book (`<Prompt1>`), since she is looking at the camera. Our **GAR-8B** manages to recognize such details while both Gemini-2.5-Pro (DeepMind, 2025b) and OpenAI-o3 (OpenAI, 2025) fail. From the last two cases, we can tell that models are still struggling with understanding complex relationships with *more than two objects*. Image source: Shao et al. (2019).

## H USE OF LLMS

In preparing this paper, LLMs are utilized as a general-purpose assistive tool. Specifically, the use of LLMs is strictly limited to proofreading the author-written text for grammatical errors, spelling

corrections, and improvements to language clarity. This application is consistent with the use of conventional grammar-checking software and did *not* extend to research ideation, data analysis, or the generation of any substantive content.

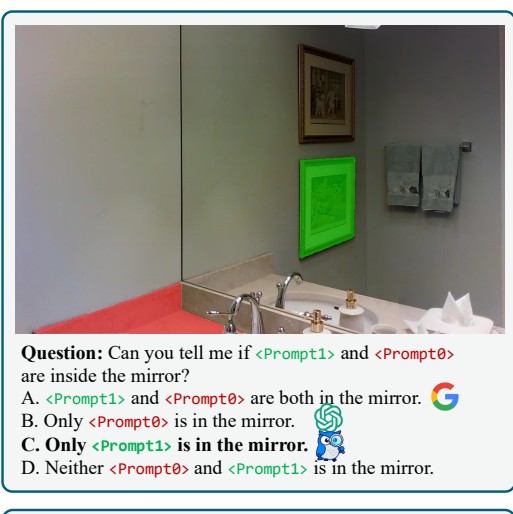

**Question:** Can you tell me if `<Prompt1>` and `<Prompt0>` are inside the mirror?
A. `<Prompt1>` and `<Prompt0>` are both in the mirror.
B. Only `<Prompt0>` is in the mirror.
**C. Only `<Prompt1>` is in the mirror.**
D. Neither `<Prompt0>` and `<Prompt1>` is in the mirror.

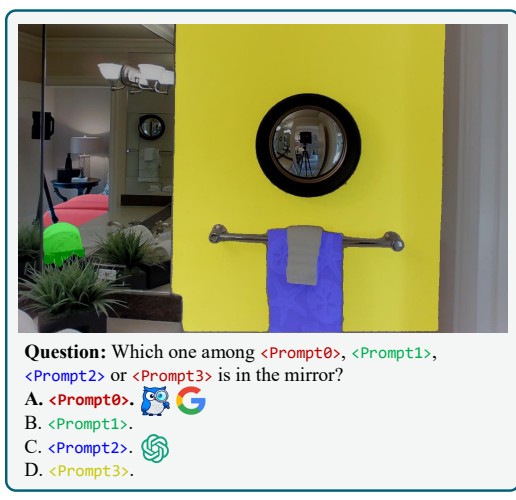

**Question:** Which one among `<Prompt0>`, `<Prompt1>`, `<Prompt2>` or `<Prompt3>` is in the mirror?
**A. `<Prompt0>`.**
B. `<Prompt1>`.
C. `<Prompt2>`.
D. `<Prompt3>`.

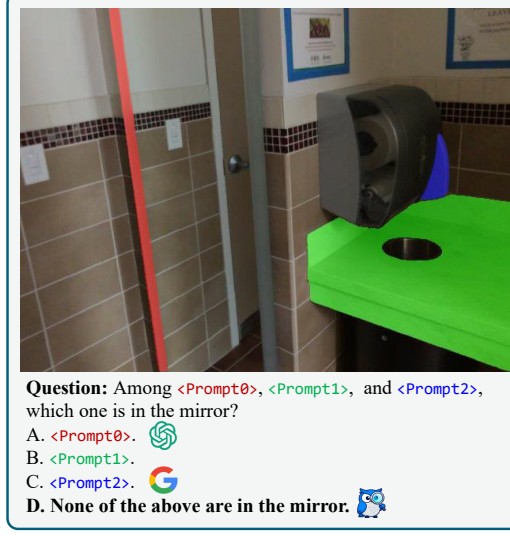

**Question:** Among `<Prompt0>`, `<Prompt1>`, and `<Prompt2>`, which one is in the mirror?
A. `<Prompt0>`.
B. `<Prompt1>`.
C. `<Prompt2>`.
**D. None of the above are in the mirror.**

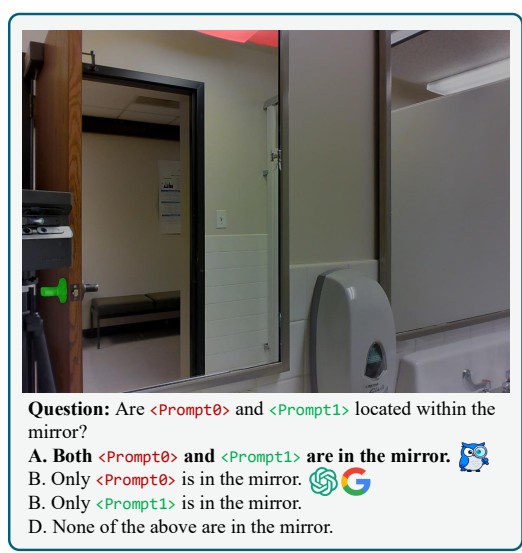

**Question:** Are `<Prompt0>` and `<Prompt1>` located within the mirror?
**A. Both `<Prompt0>` and `<Prompt1>` are in the mirror.**
B. Only `<Prompt0>` is in the mirror.
B. Only `<Prompt1>` is in the mirror.
D. None of the above are in the mirror.

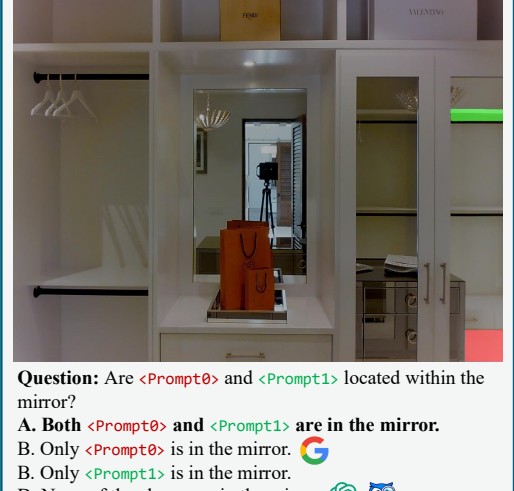

**Question:** Are `<Prompt0>` and `<Prompt1>` located within the mirror?
**A. Both `<Prompt0>` and `<Prompt1>` are in the mirror.**
B. Only `<Prompt0>` is in the mirror.
B. Only `<Prompt1>` is in the mirror.
D. None of the above are in the mirror.

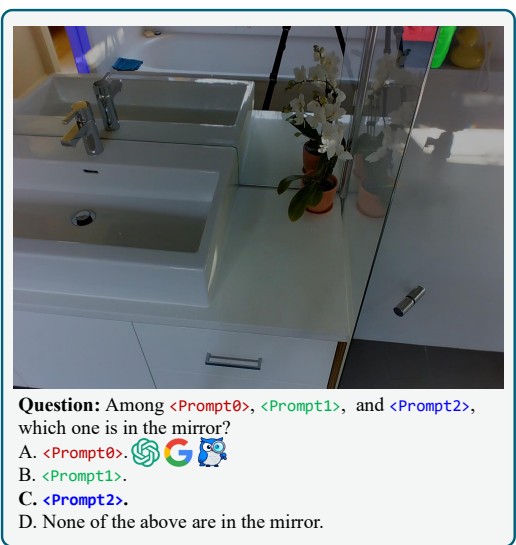

**Question:** Among `<Prompt0>`, `<Prompt1>`, and `<Prompt2>`, which one is in the mirror?
A. `<Prompt0>`.
B. `<Prompt1>`.
**C. `<Prompt2>`.**
D. None of the above are in the mirror.

Figure 9: **Qualitative comparisons** on the "non-entity recognition" protocol of our **GAR-Bench-VQA**, including two failure cases (bottom). From the last two cases, we can tell that models are sometimes still struggling with recognizing non-entities, especially distinguishing reflection from the mirror (`<Prompt2>`) and other surfaces (`<Prompt0>` and `<Prompt1>`).

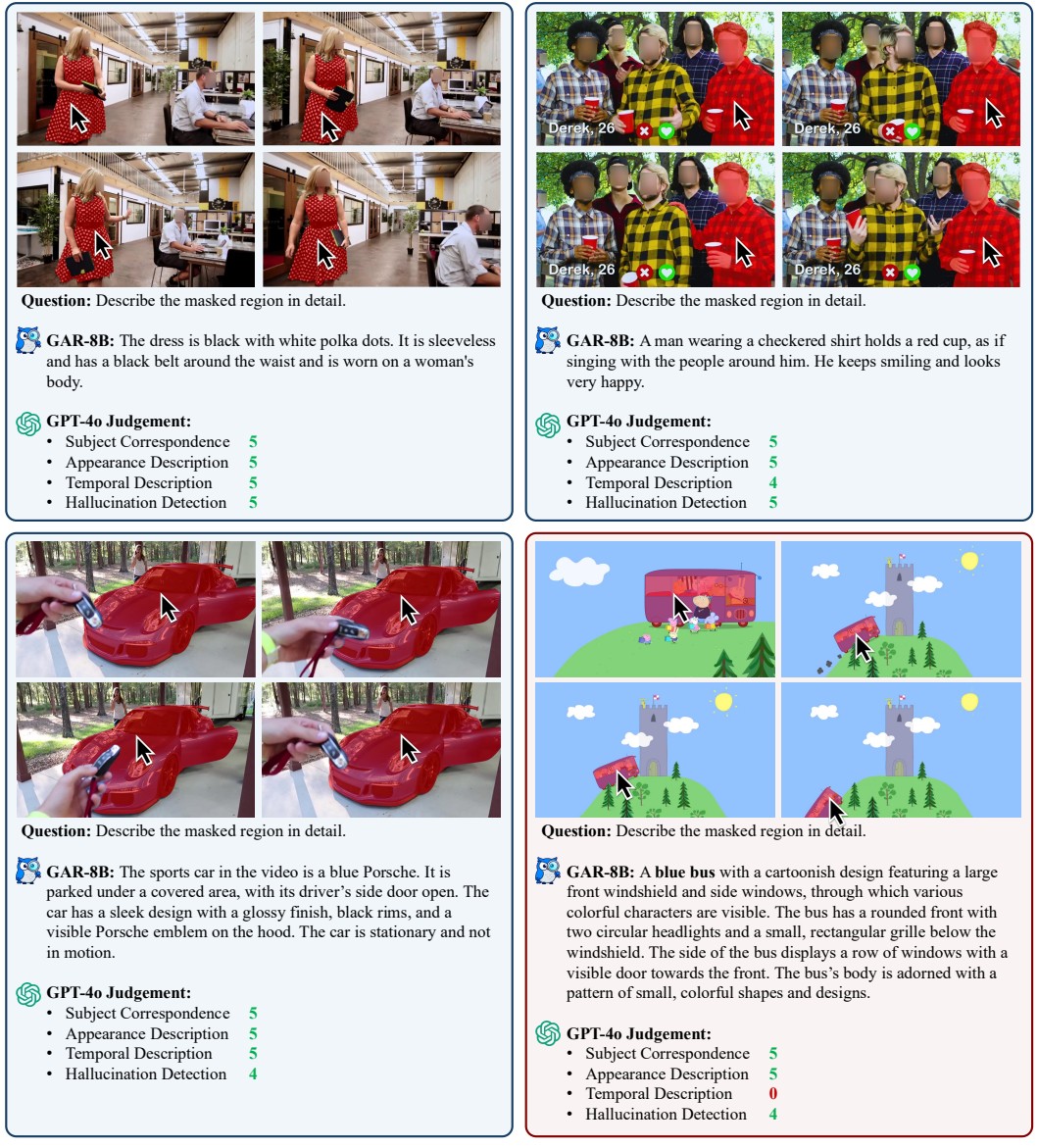

Figure 10: Qualitative results of **detailed video captioning** on VideoRefer-Bench[D] (Yuan et al., 2025b), including one failure case with a low "temporal description" score. Video source: (Chen et al., 2024b).

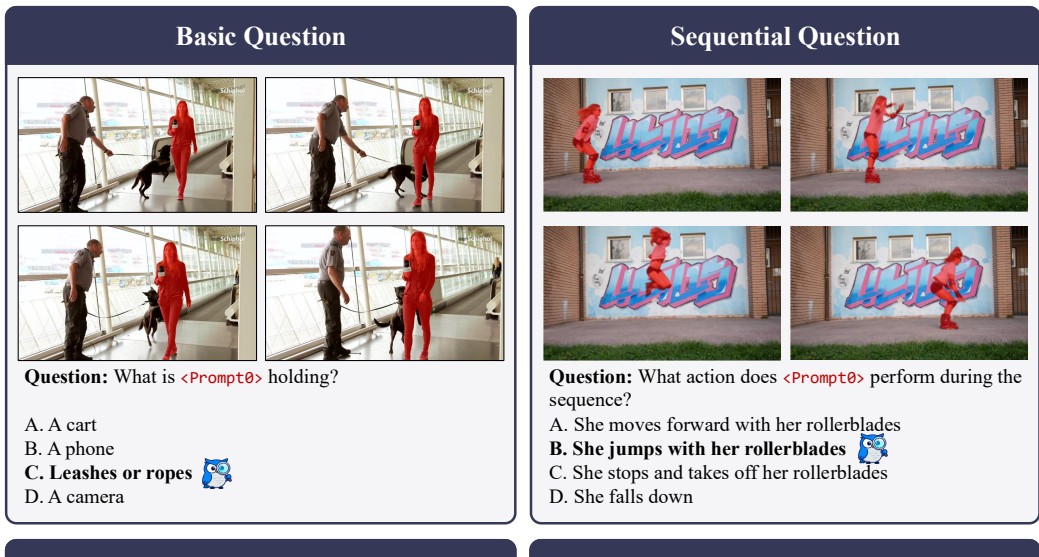

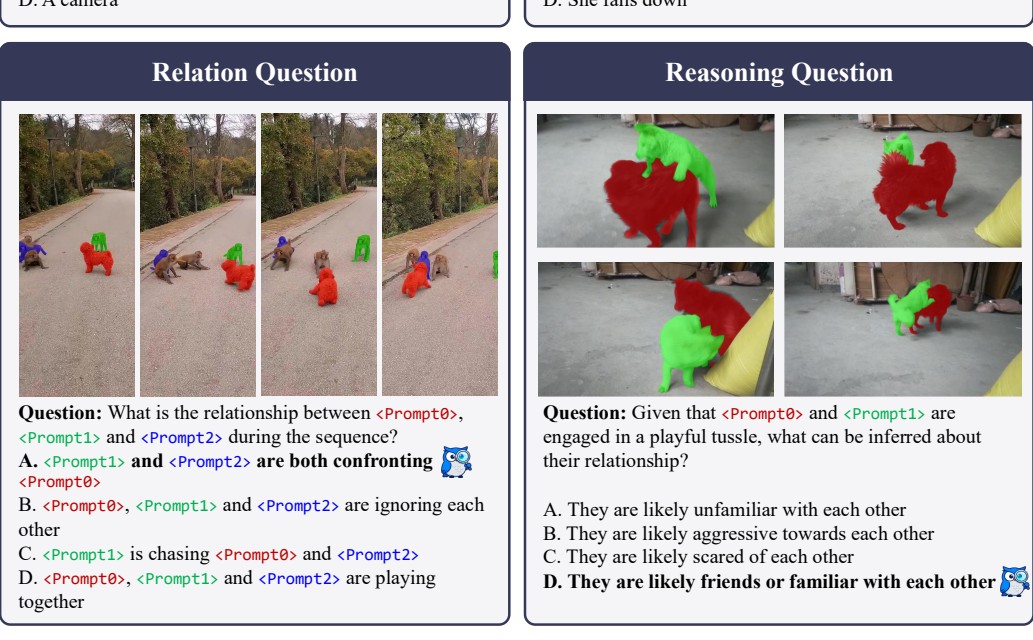

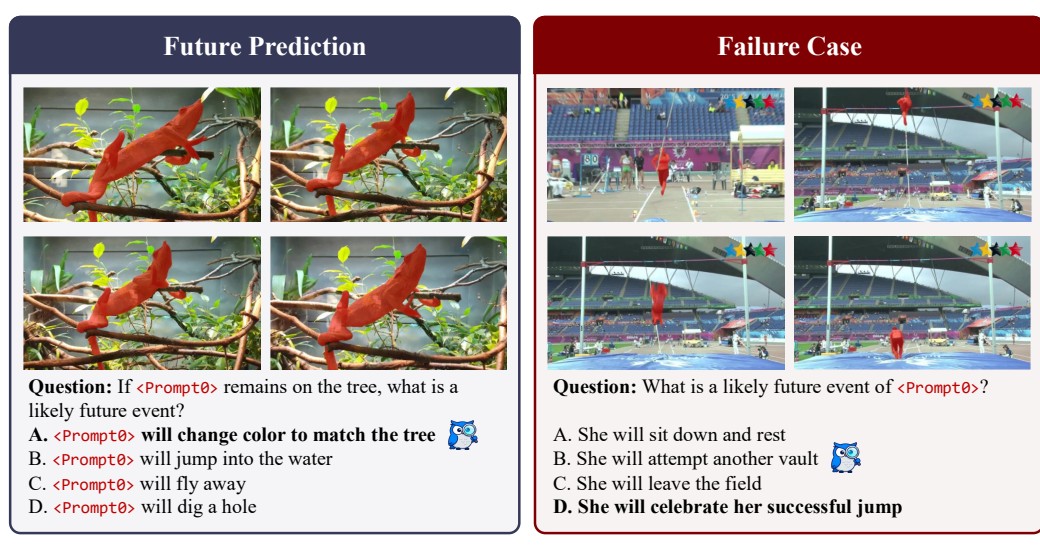

Figure 11: Qualitative results of **detailed video understanding** on VideoRefer-Bench[Q] (Yuan et al., 2025b), including one failure case in the "future prediction" protocol.

**Judge Prompt**

Answer the multiple-choice question based on the text description of an object in an image. You need to follow these rules:
1. Do not output any reasoning. Do not perform correction. Please output exactly one answer from the choices for each question. Do not repeat the question.
2. There is no need for exact matching. Please choose the closest option based on the description.

The description is: {pred_caption}

From the description above, please answer the following question with one of the choices:

Is it likely that the objects in the description are {class_name} or objects of a similar type? Again, it does not have to be an exact match.

**False Negative**

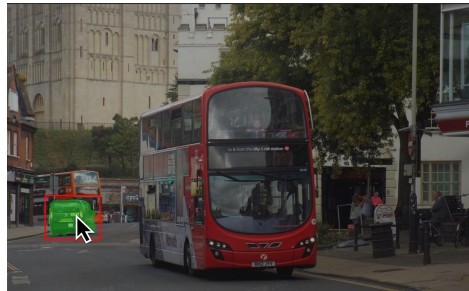

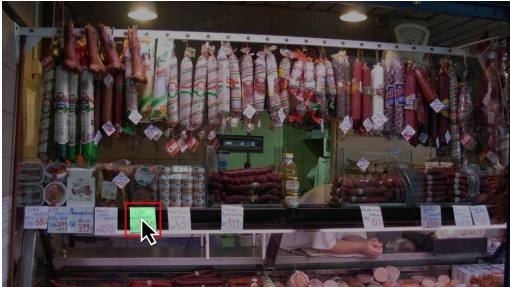

Category: Van
Description: A **black taxi** with the word TAXI in yellow on the rear window, featuring a yellow license plate and red tail lights.

GPT-4o: **Yes**

LLaMA3.1-8B: **No**

Category: Sign / Banner
Description: A **white price tag** with handwritten text in blue ink. The text reads L64 at the top, followed by 100g and 100g again below it. At the bottom, there is a large €390

GPT-4o: **Yes**

LLaMA3.1-8B: **No**

**False Positive**

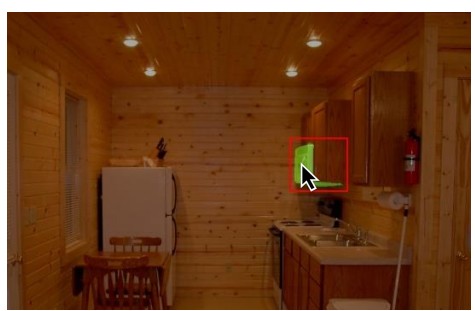

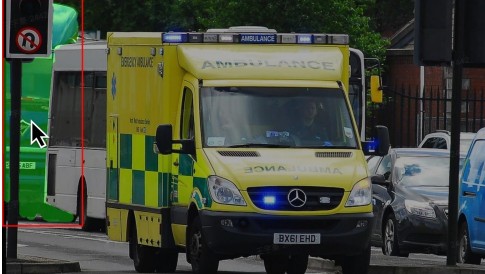

Category: Microwave
Description: A **wooden cabinet** with a vertical grain pattern, featuring a single visible door handle on the right side, which is curved and metallic. The cabinet has a flat top and appears to be part of a larger piece of furniture.

GPT-4o: **No**

LLaMA3.1-8B: **Yes**

Category: Truck
Description: The **bus** is predominantly blue with a sleek, modern design. It features a black and white logo on the side, and the word **AMBULANCE** is visible in white letters on a black background. The bus has a large, curved windshield and a side mirror extending from the front.

GPT-4o: **No**

LLaMA3.1-8B: **Yes**

Figure 12: **Incorrect *text-only* judging results** using LLaMA3.1-8B (Grattafiori et al., 2024) on DLC-Bench (Lian et al., 2025). The model is required to judge whether the description is consistent with the ground-truth category name. We illustrate both correct and wrong results. Providing extra cropped images and masks to GPT-4o (OpenAI, 2024a) effectively eliminates this issue.

---
**Prompt for Judging Descriptions and Ground-Truth Categories**

Answer the multiple-choice question based on the text description of an object in an image. You need to follow these rules:
1. Do not output any reasoning. Do not perform correction. Please output exactly one answer from the choices for each question. Do not repeat the question.
2. There is no need for exact matching. Please choose the closest option based on the description.

The description is: {pred_caption}

From the description above, please answer the following question with one of the choices:

Is it likely that the objects in the description are {class_name_list} or objects of a similar type? Again, it does not have to be an exact match.

---

Figure 13: Prompt for judging the description and the ground-truth category.

---
**Prompt for Generating Relation-Aware Captions**

You are given the following information:
    - Subject name: {subject_name}
    - Object name: {object_name}
    - Predicate (relation): {predicate_name}
    - Subject description: {sub_caption}
    - Object description: {obj_caption}

**Instructions:**
    1. First Judge if the objects in the 'Subject description' are {subject_name} or objects of a similar type.
It does not have to be an exact match. If it does not, output only: False.
    2. The 'Object description' does not need to match the 'Object name'.
       - If the 'Object description' matches the 'Object name', you may use it.
       - If it does not match, ignore it and only use the 'Object name'.
    3. Generate a fluent caption focusing mainly on the Subject.
    - Preserve as much detail from the subject description as possible.
    - Also include the relation ({predicate_name}) with the object (using either the 'Object description' if valid, or the 'Object name').
    4. Output only the final caption, without any explanations or reasoning.

---

Figure 14: Prompt for generating relation-aware caption.

---

**Prompt for Generating Question-Answering Pairs**

You are a professional Visual Question Answering (VQA) expert. Your task is to create high-quality, direct question-answer pairs about a virtual scene, based on provided ground truth data.

**Input Format:**
I will provide you with the ground truth for a scene in two JSON formats:
- captions: A dictionary containing reference tags for objects (e.g., <Prompt0>), their corresponding category (category_name), and a detailed text description (caption).
- relations: A dictionary that describes the relationships between objects using the format <subject>, <object>, <predicate>.

**Task & Output Format:**
Your task is to use this ground truth data to generate a JSON array containing 1-3 question-answer pairs.
Your output must be a single, valid JSON array and nothing else. Do not include any explanations, comments, or text outside of the JSON structure. The format should be as follows:

```json
[
  {
    "question": "The text of the question...",
    "answer": "A direct, factual answer in a short sentence or phrase."
  }
]
```

**Core Generation Rules:**
1. Core Focus on Relationships:
All questions must primarily test the spatial, action-based, or state-based relationships defined in the relations data.

2. Formulate Concise and Factual Answers:
The answer_text must directly and accurately respond to the question.
The answer must be a short, complete sentence or a descriptive phrase based only on the provided relations and captions.

Example:

Q: "What is the relationship between <Prompt0> and <Prompt1>?"
A: "<Prompt0> is on top of <Prompt1>."

3. Diverse Questioning Styles (Crucial):

Your questions must be varied. Emulate the following styles:
- Relationship/Arrangement: "What is the spatial relationship between <Prompt0> and <Prompt1>?" or "Describe the arrangement involving <Prompt1>, <Prompt2>, and <Prompt3>." or "Which statement accurately describes the positions of <Prompt0>, <Prompt2>, and <Prompt1>?"
- Comprehensive Statements: "Can you describe the arrangement involving <Prompt1>, <Prompt2> ,and <Prompt3>?"
- Location: "Where is <Prompt2> located relative to <Prompt3>?" or "How are <Prompt2> and <Prompt1> positioned relative to <Prompt0>?"
- Action & State: "What is the primary activity of <Prompt0>?" or "What are <Prompt0> doing on <Prompt4>?" or "Which statement best synthesizes the relationships involving <Prompt0> and <Prompt1>?"
- Attribute-based (using caption details): "What is on the back of the giraffe <Prompt2>?"
- Direct Relationship: "What is the spatial relationship between <Prompt0> and <Prompt1>?" or "How is <Prompt0> interacting with <Prompt1> and <Prompt2>?"
- Ask for prompt: "Which are/is described as driving on <Prompt1>?"  or "which object is located between <Prompt3> and <Prompt1>?" (the answer should be like "<PromptX>" or "<Prompt0> and <Prompt2>")
- You can vary your question from these styles or use styles not appear in here.

4. Synthesize Information for Reasoning:
Whenever possible, design questions that require synthesizing multiple relationships to arrive at the correct answer. The answer should reflect this synthesis.

5. Intelligent Use of captions:
Utilize the category_name and caption details to formulate more specific, context-aware questions and answers.

6. Strict Formatting and Wording (Crucial):
Immersive Phrasing: Frame questions as if asking about a real visual scene. Crucially, you must not use phrases like "Based on the provided relationships," or "According to the information."
Tag-Only References: You must use the <PromptX> tags to refer to objects. Do not add descriptions to the tags themselves (e.g., use <Prompt0>, not the car <Prompt0>).

**Input:**
captions: {captions}
relations: {relations}

Figure 15: Prompt for generating question-answering pairs.

**Prompt for Generating Multiple Choices Questions**

You are a professional Visual Question Answering (VQA) expert. Your task is to create high-quality, diverse, multiple-choice questions about a virtual scene based on provided ground truth data.

**Input Format:**
I will provide you with the ground truth for a scene in two JSON formats and some images:
captions: A dictionary containing reference tags for objects (e.g., <Prompt0>), their corresponding category (category_name), and a detailed text description (caption).
relations: A dictionary that describes the relationships between objects using the format
<subject>,<object>,<predicate>.
images: The Full image and mask crop images which stand for specific <PromptX>

**Task & Output Format:**
Your task is to use this ground truth data to generate a JSON array containing 1-3 multiple-choice questions.
Your output must be a single, valid JSON array and nothing else. Do not include any explanations, comments, or text outside of the JSON structure. The format should be as follows:

```json
[
  {
    "question": "The text of the question...",
    "options": ["A. ...", "B. ...", "C. ...", "D. ..."],
    "answer": "A"
  }
]
```

**Core Generation Rules:**
1. Core Focus on Relationships:
All questions must primarily test the spatial, action-based, or state-based relationships defined in the relations data.
The correct answer must be directly verifiable from the provided ground truth.

2. Diverse Questioning Styles (Crucial):
Do not overuse a single question format. Your questions must be varied. Emulate the following styles based on the examples provided in the user's file:
Comprehensive Statements: "Which of the following statements accurately describes the arrangement involving <Prompt1>, <Prompt2>, and <Prompt3>?"
Location & Belonging: "Which of the following objects are all located on(beside,on,parked on...) <Prompt4>?"
Action & State: "What is the primary activity of <Prompt0>?" or "What are <Prompt0> and <Prompt1> doing on <Prompt3>?"
Attribute-based (using caption details): "Which object, described as having an illustration of a cat, is located on <Prompt2>?" or "Which surface is the giraffe <Prompt2> lying on?"
Direct Relationship(mainly): "What is the spatial relationship between <Prompt0> and <Prompt1>?" or "How is <Prompt0> interacting with <Prompt1> and <Prompt2>?" or "Which objects are located beside <Prompt1>?",

3. Synthesize Information for Reasoning:
Whenever possible, design questions that require synthesizing multiple relationships to arrive at the correct answer. For example, a question might test <PromptA>'s relationship to both <PromptB> and <PromptC>.

4. Intelligent Use of captions:
Utilize the category_name and caption details not just for creating distractors, but to formulate more specific, nuanced, and context-aware questions and answers.

5. Plausible Distractors:
Each question must have one correct answer and 2-3 plausible but incorrect distractors.
Create these by altering the subject, object, or predicate from a correct relationship, or by using other objects from the scene to create a false but believable statement.
Use summary options like "Both <Prompt0> and <Prompt1>" or "None of the above" where appropriate.

6. Strict Formatting and Wording (Crucial):

Immersive Phrasing: Frame questions as if asking about a real visual scene. Crucially, you must not use phrases like "Based on the provided relationships," "According to the information," or reference the data sources in any way.
Tag-Only References: You must use the <PromptX> tags to refer to objects in both questions and options. Do not add descriptions to the tags themselves (e.g., use <Prompt0>, not the car <Prompt0>).
Category-Agnostic Questions: When asking "Which...", you must use general phrasing. For example, always use "Which of the following is..." instead of "Which person is..." to ensure the question remains valid for all possible answer types.

Input:
captions:{captions}
relations:{relations}

Figure 16: Prompt for generating multiple-choice questions.

