# OpenReview forum: "Grasp Any Region: Towards Precise, Contextual Pixel Understanding for Multimodal LLMs"
_ICLR.cc/2026/Conference — ICLR 2026 Poster_

### Official Review · Reviewer_ip5b · 2025-10-31

**Soundness:** 2
**Presentation:** 2
**Contribution:** 3
**Rating:** 4
**Confidence:** 4

**Summary:**

The paper proposes Grasp Any Region (GAR), a region-level MLLM that aims to combine precise region perception with access to global scene context, support interactions among multiple region prompts, and enable region-centric compositional reasoning in a single-turn conversation. The core technical idea is an RoI-aligned feature replay pipeline: the model encodes the full image with prompt masks to obtain a global, context-aware feature map, then re-pools prompt-specific features with RoI-Align so the LLM receives both prompt-aware global tokens and high-fidelity local tokens. The paper also proposes a dataset of 2.5M-scale assembled in two rounds to strengthen fine-grained recognition and relational understanding, and GARBench, a benchmark suite with a captioning protocol for multi-prompt relations and VQA that separates basic perception from reasoning such as position, non-entity recognition, and relations. The results section reports strong numbers on DLC-Bench and competitive results on GARBench, with qualitative examples

**Strengths:**

– The data contribution is good and will likely be useful to the community. The paper constructs a large training set that starts from Describe-Anything, augments with fine-grained category supervision, and then adds relation-aware descriptions and QAs from PSG using an LLM merger. This staged pipeline is clearly described with concrete numbers and roles for each round, and it targets the specific capability gaps claimed by the model, like fine-grained region recognition and multi-region relations. The ablation helps understand what each round adds.

– The benchmark contribution is also meaningful. GARBench decomposes evaluation into captioning of relations across prompts and a VQA protocol that cleanly separates perception (color, shape, material, texture) from reasoning (position, non-entity recognition, relations). The paper also uses multiple-choice designs in places where open-ended judging is brittle, and it includes tasks that force models to use global context, like mirror reflections and grid positions. Together, this looks like a well-scoped benchmark for region-aware reasoning.

**Weaknesses:**

- The major weakness is that the paper has not positioned its novelty in the literature with enough precision. The paper frames the novelty around inclusion of global context for precise perception, interactions between multiple prompts, and reasoning capabilities, but the related works section states that “previous approaches only support a single visual prompt, and often neglect global context”(L123). This statement is too strong, given that prior works referred in the same section (GPT4RoI and GLaMM) already support multiple regions and attempt to preserve global context, so the current phrasing risks overstating the novelty.

- Regarding the specific point about inclusion of global context would benefit from a clear technical comparison to GPT4RoI‑style designs.
GPT4RoI encodes the entire image once to obtain a global visual representation, then extracts RoI‑aligned features for user‑specified regions (via boxes) and interleaves those region features with the text sequence so the LLM can answer region‑centric queries while still using whole‑image context, it supports multiple regions and treats each region as an additional visual cue alongside the global image feature. The paper follows essentially the same recipe - one global image pass + RoI‑aligned region features fed to the LLM for multi‑region reasoning, with the only material differences being that (i) it takes masks rather than boxes and injects a learned mask embedding into the vision backbone before computing the global feature map (so the map itself is prompt‑aware), and (ii) instead of pooling each RoI down to a single vector, it replays a small set of RoI‑aligned tokens per region taken from that prompt‑aware map. In other words, apart from mask conditioning and unpooled multi‑token RoI replay, the overall architecture and goal, joining global context with local RoI features to enable fine‑grained, multi‑prompt understanding, are very similar.

- Comparisons currently center on DAM, but GPT4RoI is the key reference for multi-region prompting and should anchor the novelty discussion. A simple ablation in a GPT4RoI style would make the technical contribution clear, keeping the LLM and data fixed. This ablation would show whether the gains come from the architecture rather than data or evaluation choices.

- The only clear novelty that remains well supported is the region‑specific reasoning emphasis, which is indeed useful since even strong models like Gemini-2.5-Pro still fail on spatial reasoning. GARBench’s design and the strong results there are aligned with this direction, yet the novelty narrative should reflect that emphasis more, rather than attributing novelty to global context or single‑prompt limitations in prior work.

**Questions:**

Could you precisely differentiate GAR from GPT4RoI-style architectures beyond the use of RoI-Align? Please explain whether GAR’s mask changes the vision encoder’s computation to create prompt-aware global features, how this differs from methods that inject regions only at the language side after a prompt-agnostic image encoding, and what effect this has on using global context.

---

> ### Author Response · Authors · 2025-11-25
> **Response to Reviewer ip5b (Part 1)**
>
> We sincerely thank you for your insightful and expert review. Your comments are invaluable and have helped us significantly clarify the core contributions of our work. You correctly pointed out that our manuscript did not sufficiently differentiate our method from prior works like GPT4RoI, which led to valid concerns about our novelty. We apologize for this lack of clarity. Our initial draft's emphasis on contrasting with more recent methods like DAM inadvertently obscured the more fundamental architectural differences from the GPT4RoI-style paradigm.
>
> We would like to clarify our core motivation, which is to address a fundamental trade-off in region-level MLLMs: **effectively balancing global scene context with fine-grained local details**. We observe that existing methods struggle to master both. For instance, methods like DAM excel at local perception but lack a holistic view of the global context. Conversely, earlier works like GPT4RoI and GLaMM, while incorporating the global image, tend to lose crucial local details by pooling region features into single vectors. Their weak performance on fine-grained benchmarks like DLC-Bench supports this observation.
>
> **Our proposed GAR architecture is designed specifically to solve this dilemma.** The key novelty lies not just in using RoI-Align, but in a unique RoI-aligned feature *replay* pipeline that *inherently preserves both global and local information*. Moreover, and in direct contrast to GPT4RoI's prompt-agnostic image encoding, our method injects mask embeddings into the vision backbone. This makes the entire global feature map prompt-aware before any region features are extracted. The subsequent replay of high-resolution local tokens from this context-aware map ensures that the LLM receives both high-fidelity local information and a global context that is already attuned to the specific region(s) of interest. *Empirical comparison between GAR-like methods and GPT4RoI-like approaches can be found in **A3**.*
>
> We believe this architectural distinction is the key to our model's strong performance and directly addresses the novelty concerns you raised. Thanks again for correctly pointing out the inappropriate literature review, and *we have tried our best to rewrite both the introduction and the related work section in our revised version.*
>
> Detailed point-to-point responses are provided below.
>
> **W1: About the related work section.**
>
> **A1**: We sincerely thank you for this crucial comment. You are right. The statement in L123 is imprecise and overstates our novelty by inaccurately characterizing prior work like GPT4RoI and GLaMM. We apologize for this oversimplification and have thoroughly revised the manuscript to provide a more precise and fair comparison.
>
> Our intention was not to claim that these models completely neglect global context, but rather to highlight a critical trade-off they face. As we briefly alluded to in our introduction, the core challenge is to *simultaneously leverage global context while preserving sufficient local details (L51-53 in our original version)*.
>
> This is precisely where our work diverges from the GPT4RoI-style paradigm. While these methods do process the full image, they typically distill each region into a single pooled feature vector before feeding it to the LLM. This aggressive downsampling inevitably leads to a significant loss of fine-grained information, leading to their weak performance on detail-intensive benchmarks like DLC-Bench.
>
> Our work directly tackles this limitation. By introducing the RoI-aligned feature replay pipeline, we preserve a set of high-fidelity, unpooled local tokens from a prompt-aware global feature map. This ensures that the model can reason about a specific region with rich local details, while the global context it relies on is already attuned to that region of interest.
>
> We are grateful for your feedback, which has pushed us to clarify this key distinction. In the revised version, we have replaced the inaccurate statement and restructured our related works to frame our contribution.

---

> ### Author Response · Authors · 2025-11-25
> **Response to Reviewer ip5b (Part 2)**
>
> **W2 & W3: Comparison with GPT4RoI.**
>
> **A2 & A3**: We sincerely thank you for this excellent and constructive suggestion. You have precisely identified the most critical comparison needed to isolate our architectural contributions. We agree that a direct, fair ablation against a GPT4RoI-style baseline is the best way to demonstrate the value of our proposed architecture.
>
> Following your advice, we conducted a new ablation study, keeping the MLLM, training data, and training schedule fixed. We implemented a "GPT4RoI-like" baseline that mirrors its core design principles:
> - Global: It uses a prompt-agnostic vision encoder (i.e., no mask conditioning) to generate a global feature map.
> - Local: It extracts a single, RoI-pooled feature vector for each region, mimicking the information-compression step in GPT4RoI and similar methods.
> This creates a minimal pair comparison against our "GAR-like" architecture, which features:
> - Global: A prompt-aware global feature map via "Image + mask" conditioning.
> - Local: Rich, unpooled local details via "RoI-aligned feature replay".
> The results, evaluated on both our GARBench and the fine-grained DLC-Bench, are presented below:
>
> | Note | Global | Local | GARBench-Cap | GARBench-VQA | DLC-Bench |
> |---|---|---|---|---|---|
> | GAR-like | Image + mask | RoI-aligned feature replay | 57.5 | 50.6 | 77.1 |
> | GPT4RoI-like | Image | RoI-pooled feature | 39.3 | 39.1 | 50.0 |
>
> Our GAR-like architecture substantially outperforms the GPT4RoI-like baseline across all benchmarks, with a staggering +27.1 point gain on the detail-oriented DLC-Bench (77.1 vs. 50.0).
>
> This significant performance gap directly validates our core thesis and demonstrates that the architectural differences you identified are indeed material. The two key advantages of our design are:
> - **"RoI-aligned feature replay" is critical for preserving the sufficient local details** needed for fine-grained perception tasks, which are lost in the RoI-pooling step of the baseline. This explains the massive lead on DLC-Bench.
> - **"Mask conditioning" creates a *prompt-aware* global context**, allowing the model to better perform compositional reasoning and understand inter-region relationships, leading to significant gains of +18.2 on GARBench-Cap and +11.5 on GARBench-VQA.
> In summary, this ablation confirms that our architectural innovations, rather than just data or evaluation choices, are the primary drivers of GAR's superior performance. We have added this crucial ablation study to the 5th row in Table 8. We are very grateful for this suggestion, as it has substantially strengthened our paper.
>
> **W4: About the emphasis direction.**
>
> **A4**: Thank you for this insightful comment. We are greatly encouraged that you find our emphasis on region-specific reasoning and the design of GARBench to be a clear and useful contribution. You have given us a very clear direction to refine our paper's narrative, and we fully agree that we should pivot to highlight this strength more prominently.
>
> We would like to clarify how we see the connection between our core motivation and this capability. We posit that **strong region-specific reasoning is the direct and most significant outcome of successfully resolving the fundamental trade-off between global context and local details.**
>
> Sophisticated reasoning, such as understanding spatial positions, identifying non-entity elements like reflections, or deducing inter-object relations, is exceptionally challenging precisely because it requires the model to simultaneously:
>
> - **What** it is: Perceive the fine-grained, intrinsic properties of a specific region.
> - **Where** it is and **why** it matters: Understand its context, position, and relationship to the broader scene.
>
> A model that lacks local details (like GPT4RoI and GLaMM) might misidentify the object, while a model that lacks global context (like DAM) cannot reason about its position or relationships.
>
> Therefore, we design GAR to specifically overcome this trade-off, preserving rich local details via feature replay from a prompt-aware global map. We provide the model with the essential visual foundation required for such complex reasoning. The strong performance on GARBench is, in essence, the empirical validation that our architectural solution to the trade-off problem has successfully enabled these advanced reasoning capabilities.
>
> Following your excellent advice, we have revised the paper's narrative in the introduction. We have reframed our primary contribution, while presenting our architectural solution to the global-local dilemma as the core technical enabler of this capability.

---

### Official Review · Reviewer_n7Lw · 2025-11-01

**Soundness:** 3
**Presentation:** 3
**Contribution:** 3
**Rating:** 6
**Confidence:** 4

**Summary:**

This paper proposes a vision-language model that enhances region-level visual understanding by combining global context with fine-grained local detail. Using a RoI-aligned feature replay mechanism, GAR processes full-image features while focusing on specific regions, enabling accurate reasoning about multiple prompts (e.g., spatial and relational understanding). The authors also introduce GARBench, a new benchmark that evaluates both single-region captioning and multi-region reasoning. Experiments show GAR surpasses previous models like DAM-3B, Ferret, and even larger models such as InternVL3-78B on both captioning and VQA tasks.

**Strengths:**

+ Clear motivation and reasonable architectural solution:
GAR tackles the important problem to understand regional information in the vision-language model, and proposes the reasonable solution with a RoI-aligned feature replay strategy. This design allows global context retention while focusing on high-resolution local features.

+ Focus on inter-region relationships:
Unlike prior works, which primarily handle single-object or localized descriptions, GAR explicitly models multi-prompt interactions such as spatial relations and entity reflection recognition. This makes it suitable for compositional reasoning tasks previously underexplored by region-level MLLMs

+ Comprehensive benchmark contribution (GARBench):
I appreciate the GAR Benchmark contribution. It evaluates not only region captioning but also relational VQA, multi-prompt reasoning, and non-entity discrimination—offering a more complete view of region-level comprehension capabilities

**Weaknesses:**

- Evaluation dependence on LLM judges:
The authors rely on LLM-based evaluators (e.g., GPT-4) for qualitative assessment. This introduces potential bias due to stylistic or verbosity differences among models. Cross-validation with human ratings or standardized metrics would strengthen the claims.

- Insufficient data transparency:
The training pipeline involves multiple stages—seed captioner, LLM merger, and relational caption generation—but lacks detail on data deduplication and leakage checks. Since datasets like COCO or PSG may overlap with benchmarks, more rigorous data hygiene reporting is necessary

- Marginal novelty compared to related works:
The combination of RoI-Align with contextual replay is technically effective but not radically novel; similar ideas of multi-scale or global-local fusion appear in Ferret and GPT4RoI.

- Missing references: The paper misses recent region-understanding and grounding works such as [ref1] and [ref2], and would benefit from a more detailed comparison and discussion of how GAR differs from these grounded vision-language models.

[ref1] Toward Interactive Regional Understanding in Vision-Large Language Models, NAACL 2024

[ref2] Groma: Grounded Multimodal Assistant, ECCV 2024

**Questions:**

- Scalability to the number of regions:
The paper shows results for up to a few regions. What happens when the number of regions scales to tens or hundreds—does reasoning complexity or memory usage grow linearly?

- Judge consistency: How LLM judges' results are consistent? Are they consistent enough for multiple runs?

---

> ### Author Response · Authors · 2025-11-25
> **Response to Reviewer n7Lw (Part 1)**
>
> We thank Reviewer n7Lw for the thoughtful and constructive review. We are encouraged that the reviewer finds our work to have "clear motivation" and a "reasonable architectural solution" for this "important problem". We particularly appreciate that the reviewer highlights our focus on inter-region relationships, noting it "makes it suitable for compositional reasoning tasks previously underexplored". We are also grateful for the appreciation of our "comprehensive benchmark contribution" (GAR-Bench), which offers a "more complete view" of region-level capabilities.
>
> **W1 & Q2: About LLM judges.**
>
> **A1**: Thank you for this comment regarding the potential biases of LLM-based evaluators and their consistency. This is a crucial point for robust MLLM evaluation, and we appreciate you pushing us to strengthen our claims.
>
> **(1) About Judge Consistency**: To directly investigate the consistency and potential bias of the LLM judge, we conducted a new cross-judge validation experiment. We re-evaluated all models on GARBench-Cap using four different powerful LLMs as judges: GPT-4o (our original judge), o3, Gemini-2.5-Flash, and Gemini-2.5-Pro.
>
> **The results, presented in the table below, demonstrate a high degree of consistency in model rankings:**
>
> |  | Judge: GPT-4o |  | Judge: o3 |  | Judge: Gemini-2.5-Flash |  | Judge: Gemini-2.5-Pro |  |
> |---|---|---|---|---|---|---|---|---|
> | Model | GARBench-Cap | Rank | GARBench-Cap | Rank | GARBench-Cap | Rank | GARBench-Cap | Rank |
> | DAM-3B | 13.1 | 10 | 9.3 | 10 | 9.4 | 10 | 0.3 | 10 |
> | PAM-3B | 21.1 | 9 | 21.5 | 9 | 25.3 | 9 | 31.3 | 9 |
> | VP-SPHINX-13B | 32.3 | 8 | 27.5 | 8 | 30.4 | 8 | 32.7 | 8 |
> | Qwen2.5-VL-32B | 36.8 | 7 | 28.1 | 7 | 37.4 | 7 | 41.8 | 7 |
> | InternVL3-38B | 45.1 | 6 | 30.8 | 6 | 41.9 | 6 | 47.4 | 6 |
> | GPT-4o | 51.5 | 5 | 31.8 | 5 | 46.3 | 5 | 50.6 | 5 |
> | o3 | 56.9 | 4 | 37.8 | 3 | 52.8 | 4 | 60.7 | 3 |
> | GAR-1B | 57.5 | 3 | 37.4 | 4 | 55.1 | 3 | 59.8 | 4 |
> | Gemini-2.5-Pro | 59.3 | 2 | 44.5 | 2 | 57.4 | 2 | 61.3 | 2 |
> | GAR-8B | 62.2 | 1 | 46.7 | 1 | 60.3 | 1 | 62.6 | 1 |
>
> As shown, while the absolute scores fluctuate across different judges, which reflects their inherent stylistic preferences or scoring strictness, the relative ranking of models is remarkably stable. Most importantly, our GAR-8B model consistently ranks 1st across all four distinct judges, and our other GAR models also consistently place in the top tier. This new experiment strongly suggests that our primary claims about GAR's superior performance are robust and not an artifact of a single judge's bias.
>
> *We have added this analysis to **Table 14** in our revised version.*
>
> **(2) On Dependence on LLM Judges**: We agree with the reviewer that human evaluation remains meaningful. Given the time constraints of the rebuttal, we performed the cross-judge validation above as a rigorous and feasible alternative. We will perform comprehensive human validation in the future.
>
> At the same time, our methodology aligns with common practices for large-scale evaluation in the field. As the reviewer knows, recent benchmarks such as Ferret-Bench and MDVP-Bench have also adopted powerful LLMs as scalable evaluators (GPT-4V for Ferret-Bench and GPT-4o for MDVP-Bench). Our added cross-judge analysis provides an additional aspect that we believe strengthens our results.
>
> Thank you again for this constructive suggestion. We believe this new analysis significantly reinforces the validity of our evaluation.
>
> **W2: About data leakage.**
>
> **A2**: To ensure there was no data leakage between our training and evaluation sets, we actually implemented a multi-step data hygiene process:
>
> **(1) Strict Adherence to Official Splits**: For all standard benchmarks used (e.g., COCO, PSG, RefCOCO), we strictly followed the official training, validation, and testing splits. Our training data, including the initial data for the seed captioner and the LLM merger, was constructed exclusively from the designated "train" splits.
>
> **(2) Clean Benchmark Design**: As a further safeguard, we intentionally designed our GARBench to include data from sources with minimal overlap with common pre-training corpora. As mentioned in Appendix B.1, the inclusion of images from FSC-147, which does not appear in standard large-scale training sets, provides a guaranteed clean and out-of-distribution testbed.
> We are confident that this multi-step procedure prevents data leakage and ensures the fairness of our evaluation.

---

> ### Author Response · Authors · 2025-11-25
> **Response to Reviewer n7Lw (Part 2)**
>
> **W3: Comparison with Ferret and GPT4RoI.**
>
> **A3**: We would like to clarify that our novelty lies *not* in the general idea of fusion, but in the specific mechanism we propose to resolve a critical trade-off that prior methods compromise on.
>
> **(1) The Conceptual Difference: Preservation vs. Aggregation.** The core challenge is to provide both rich, global context and high-fidelity, fine-grained local details. While methods like Ferret and GPT4RoI approach this, they rely on RoI-pooling, which is an aggregation operation. It compresses all the features within a region into a single, fixed-size vector. This process inevitably leads to a loss of information, discarding the very local details needed for fine-grained recognition and reasoning.
>
> Our key insight is that this information loss is unnecessary. Our RoI-aligned feature replay is fundamentally different. It is a preservation and selection mechanism. It doesn't aggregate or compress. Instead, it directly selects and "replays" the original, high-resolution features from the vision backbone that fall within the RoI. This elegantly preserves 100% of the local detail while ensuring the features are still contextualized from being processed in the full image.
>
> **(2) Empirical Validation.** To provide concrete evidence for our claim, we conducted a controlled experiment comparing our approach to a GPT4RoI-like model that uses RoI-pooling. The results below isolate the impact of this single design choice:
>
> | Note | Global | Local | GARBench-Cap | GARBench-VQA | DLC-Bench |
> |---|---|---|---|---|---|
> | GAR-like | Image + mask | RoI-aligned feature replay | 57.5 | 50.6 | 77.1 |
> | GPT4RoI/Ferret-like | Image | RoI-pooled feature | 39.3 | 39.1 | 50.0 |
>
> The model using RoI-pooling suffers a massive performance drop across all benchmarks, particularly on DLC-Bench (50.0 vs. 77.1), a benchmark that explicitly requires understanding fine-grained local details. This result empirically confirms our hypothesis: feature pooling creates a critical information bottleneck that our feature replay method successfully avoids.
>
> *We have added this experiment to **the 5th row in Table 8** in our revised version.*
>
> In summary, while preserving global context seems to be similar, our novelty lies in proposing a simple yet highly effective solution that provides both global context and lossless local features. We have added the introduction and the related work section to make this clearer.
>
> **W4: Comparison with [ref1, ref2].**
>
> **A4**: We provide a brief discussion here. All three papers address a critical limitation of mainstream MLLMs: insufficient fine-grained regional understanding. However, model capabilities and scenarios are quite different:
> - RegionVLM [ref1]: Excels at interactive single-region captioning and zero-shot referring segmentation but struggles with context-dependent region interpretation and multi-region reasoning.
> - Groma [ref2]: Strong at precise visual grounding and grounded dialogue, with superior localization on small objects (via DINOv2) but limited to box-level interactions (no pixel-level mask support).
> - GAR: Unifies three core capabilities: precise single-region description, multi-region relationship modeling, and advanced compositional reasoning. It also supports zero-shot video transfer and even outperforms larger models (e.g., InternVL3-78B) on multi-region tasks.
> GAR differentiates itself by addressing two critical gaps in RegionVLM and Groma:
> 1. Context-Aware Regional Analysis: Unlike RegionVLM's text-based spatial encoding and Groma's isolated region tokens, GAR's RoI-aligned feature replay inherently merges local details and global context, eliminating misinterpretation of region semantics (e.g., non-entity recognition).
> 2. Multi-Region Reasoning: While RegionVLM and Groma focus on single/multi-region alignment, GAR prioritizes compositional reasoning across arbitrary numbers of regions, enabled by its relation-aware training data and GAR-Bench's dedicated evaluation of interactions.
> 3. Versatility: GAR's zero-shot video transfer and performance on both region-specific and general multimodal benchmarks (e.g., MMVP, RealWorldQA) demonstrate broader applicability than RegionVLM (limited to static regions) and Groma (focused on grounding).
> We have added these two approaches to the related work section.
>
> [ref1] Toward Interactive Regional Understanding in Vision-Large Language Models, NAACL 2024
>
> [ref2] Groma: Grounded Multimodal Assistant, ECCV 2024

---

### Official Review · Reviewer_SknR · 2025-11-02

**Soundness:** 3
**Presentation:** 3
**Contribution:** 2
**Rating:** 4
**Confidence:** 4

**Summary:**

This paper aims to enhance the capability of Vision-Language Models (VLMs) in understanding dense visual scenes, particularly in capturing fine-grained local details and the relationships among multiple local regions. The authors propose the GAR method, whose core lies in the ROI-aligned feature replay technique, which enhances local visual feature representation through ROI alignment during the visual feature extraction process. Furthermore, the paper introduces a large-scale GAR-2.5M training dataset and GARBench benchmark for training and evaluating the GAR-1B/8B models, respectively. Experimental results across multiple datasets demonstrate the superior performance of the proposed GAR-1B and GAR-8B models.

**Strengths:**

- Understanding fine-grained details and object inter-relationships is critical for real-world applications of VLMs. This paper provides important contributions in training dataset, evaluation benchmark and model architechture.
- The paper is clearly written and easy to read.

**Weaknesses:**

- Understanding the dense world is an important capability. However, the proposed GAR task represents only a specific task formulation within this direction. In particular, the paper constrains the use of masks as indicators of objects, which may introduce bias when evaluating a model’s true understanding of dense visual scenes. The compared models might simply lack the ability to interpret masks, rather than being genuinely deficient in understanding local details and relationships.
- The proposed method in this paper is designed for a specific task type, incorporating a task-specific module — the ROI-Aligned Feature Replay Module. Moreover, the corresponding training dataset GAR-2.5M is constructed using the same methodology as GARBench, which weakens the assessment of the method’s effectiveness and generalizability. In addition to comparisons with mainstream open-source models such as Qwen2.5-VL and InternVL3 on GARBench, evaluations on general benchmarks would further strengthen the validation of the proposed method and dataset in terms of their effectiveness and general applicability.
- The ablation study in this paper is not sufficiently comprehensive. In particular, it would be beneficial to include an additional variant that uses only global patch embeddings and mask embeddings, without the ROI-Aligned Feature Replay Module, and train this variant on the GAR-2.5M dataset. This result would help highlight the effectiveness of the ROI-Aligned Feature Replay Module. Furthermore, comparisons with the pretrained model, namely the PerceptionLM baseline, should also be added across all datasets to provide a more complete evaluation.
- The implementation details of GAR are not clearly described. In particular, how are the RoI-Aligned features integrated with the global context features and question tokens?

**Questions:**

See Weaknesses.

---

> ### Author Response · Authors · 2025-11-25
> **Response to Reviewer SknR (Part 1)**
>
> We thank Reviewer SknR for the valuable feedback. We are happy that the reviewer agrees the problem we address is "critical for real-world applications" and acknowledges our "important contributions" across the training dataset, evaluation benchmark, and model architecture. We are also encouraged that the reviewer finds the paper "clearly written and easy to read".
>
> **W1: Current general models lack the ability to interpret masks instead of being genuinely deficient in understanding local details and relationships.**
>
> **A1**: We sincerely thank the reviewer for this insightful comment regarding the specificity of our task formulation and the potential bias introduced by using masks.
>
> **(1) The challenge is fundamental, not format-specific.** We agree that GAR represents a specific task formulation. However, we argue that it targets a foundational yet *unsolved* challenge in VLMs: *precisely grounding and reasoning about specified local regions.*
> To directly investigate the reviewer's valid concern that models may simply "lack the ability to interpret masks", we conducted a new, extensive study on leading VLMs (including GPT-4o, Gemini-2.5-Pro, and o3). We evaluated their performance using four different region-specification formats, as illustrated in **Figure 7 in our revised version**:
> - Type 1: Separated global image and local masks (our original setting).
> - Type 2: Drawing mask outlines directly onto the image.
> - Type 3: Using bounding boxes derived from masks.
> - Type 4: Using center-point crosses derived from masks.
>
> |  | GPT-4o |  |  | o3 |  |  | Gemini-2.5-Pro |  |  |
> |---|---|---|---|---|---|---|---|---|---|
> | Input Type | GAR-Bench-VQA | GAR-Bench-Cap | DLC-Bench | GAR-Bench-VQA | GAR-Bench-Cap | DLC-Bench | GAR-Bench-VQA | GAR-Bench-Cap | DLC-Bench |
> | Type 1 | 53.5 | 51.5 | 41.0 | 61.3 | 56.9 | 48.0 | 64.2 | 59.3 | 48.4 |
> | Type 2 | 56.4 | 52.0 | 38.4 | 63.4 | 54.9 | 47.8 | 61.9 | 58.1 | 52.9 |
> | Type 3 | 51.2 | 31.9 | 25.4 | 57.8 | 40.7 | 34.9 | 61.9 | 61.3 | 54.6 |
> | Type 4 | 48.3 | 38.2 | 28.5 | 59.0 | 47.1 | 41.3 | 66.0 | 44.1 | 47.8 |
>
> New experimental results demonstrate a general problem. They show that these powerful, **general-purpose VLMs consistently struggle across all four settings**. No single format is a simple "fix". For instance, even with visually intuitive formats like bounding boxes (Type 3) or outlines (Type 2), the models' performance remains far from satisfactory.
>
> This crucial finding strongly suggests that the performance gap does not stem from an inability to parse a specific input format (i.e., binary masks). Instead, it reveals a more fundamental deficiency in current VLMs' capabilities for fine-grained perception and relational reasoning, regardless of how the regions of interest are indicated.
>
> From these results, we obtain two conclusions:
> - First, the ability to "interpret a mask" is not a trivial shortcut but is itself a challenging, fine-grained understanding task that even SOTA models have not mastered.
> - Second, among the various formats, masks offer the highest precision and flexibility to delineate arbitrary, non-rectangular objects, which is crucial for complex, real-world dense scenes where bounding boxes or points are often ambiguous or insufficient.
>
> In summary, our new experiments validate that the GAR task, while specific, effectively isolates a widespread and fundamental weakness in modern VLMs. By establishing a precise and challenging benchmark, we believe our work provides a clear and valuable path toward developing more capable models for dense scene understanding.
>
> *We have added these new experiments in **Table 15 in our revised version**.*

---

> ### Author Response · Authors · 2025-11-25
> **Response to Reviewer SknR (Part 2)**
>
> **W2: About generalizability.**
>
> **A2**: We thank the reviewer for these critical questions regarding the method's generalizability and the training/evaluation setup. We address these two points below.
>
> **(1) The benefits of this training are *not* limited to our GAR-Bench.**
>
> We would like to clarify that the benefits of this training are not confined to GAR-Bench. As shown in our main results (**Tables 1-6**), our GAR models, trained on GAR-2.5M, achieve significant improvements on a wide range of other public, third-party fine-grained benchmarks, including DLC-Bench, PACO, LVIS, MDVP-Bench, Ferret-Bench, and VideoRefer. *This demonstrates that our method and data teach a **transferable skill** of fine-grained perception and reasoning, rather than merely overfitting to our own benchmark's format.*
>
> **(2) On the Performance of General VQA Benchmarks.**
>
> We appreciate the suggestion to evaluate on general benchmarks, which is crucial for assessing the broader impact of our method. We included these results in **Table 11 in our revised version**.
>
> | Method | V* | MMVP | RealWorldQA | MMStar |
> |---|---|---|---|---|
> | DAM-3B | 45.0 | 60.7 | 54.3 | 39.7 |
> | PAM-3B | 1.4 | 4.3 | 1.7 | 2.7 |
> | PerceptionLM-8B | 69.1 | 76.0 | 75.0 | 57.1 |
> | GAR-8B | 59.2 | 78.0 | 58.7 | 43.9 |
> | GAR-8B (w/ 600K LLaVA-OV-Data) | 62.3 | 79.7 | 61.8 | 51.6 |
>
> A performance drop is observed when comparing our specialist GAR-8B model directly against its generalist PerceptionLM-8B backbone. This trade-off, often termed catastrophic forgetting, is a known challenge when fine-tuning a model on specialized data.
> However, we would like to highlight two important findings:
> - **Superiority among Specialists**: When compared to other region-understanding specialist models like DAM-3B and PAM-3B, our GAR-8B demonstrates a far superior preservation of general abilities, indicating a better balance between specialization and generalization.
> - **Mitigating the Trade-off**: More importantly, we demonstrate that this trade-off is not a fundamental limitation of our approach. By training an additional variant, GAR-8B (w/ 600K LLaVA-OV-Data), which simply mixes our GAR-2.5M data with 600K general-purpose dataset sampled from LLaVA-OneVision, we can substantially recover the general VQA performance. For instance, the RealWorldQA score improves from 58.7 to 61.83, and the MMVP score improves from 78.0 to 79.7, bringing it much closer to the base model's performance while retaining its strong specialized skills.
>
> In conclusion, these results show that the specialized capabilities imparted by our method are not fundamentally at odds with general VQA abilities. The performance trade-off is manageable and can be effectively addressed with standard data-mixing strategies, confirming the general applicability and value of our contribution.
>
> **W3: About the baseline that uses only global patch embeddings and mask embeddings.**
>
> **A3**: We sincerely thank the reviewer for this suggestion to make our ablation study more comprehensive. Following the advice, we trained an additional variant that uses only global patch embeddings and mask embeddings, trained on GAR-2.5M but without the RoI-Aligned Feature Replay Module. We have also included the PerceptionLM baseline performance on all reported datasets for a complete evaluation.
>
> | Model | GARBench-Cap | GARBench-VQA | DLC-Bench |
> |---|---|---|---|
> | PerceptionLM (Baseline) | 18.2 | 37.9 | 38.4 |
> | w/o RoI-Replay (Trained on GAR-2.5M) | 57.5 | 50.6 | 77.1 |
> | Full GAR Model (w/ RoI-Replay) | 32.7 | 40.4 | 61.7 |
>
> Comparing our full model to the variant without the module, we observe a dramatic performance drop across all benchmarks. This stark contrast unequivocally demonstrates that the ROI-Aligned Feature Replay Module is the key component responsible for the superior fine-grained understanding, and the performance gain is not merely from using mask inputs.
>
> *We have integrated this more comprehensive analysis into **Table 8 of our revised paper**.*

---

> ### Author Response · Authors · 2025-11-25
> **Response to Reviewer SknR (Part 3)**
>
> **W4: About the implementation details of GAR.**
>
> **A4**: We provide the following step-by-step explanation of the feature integration process based on Figure 3 in our original manuscript.
>   - The user's "Question" contains region placeholders (e.g., `<Prompt0>`), which are associated with bounding boxes converted from its mask (e.g., `<Box0>`).
>   - The "RoI-Align" operation is performed on the intermediate feature maps from the visual encoder. It uses the provided bounding boxes to pool features corresponding precisely to each region of interest.
>   - These pooled, high-resolution local features are then processed by our `RoI-Aligned Feature Replay` module to generate a sequence of local region features (represented by the red, green, and blue tokens).
>
> The final input sequence for the LLM is constructed by concatenating these different components in a specific order.
>   - First, the global context features are placed at the beginning of the sequence.
>   - Next, the tokenized question text is appended.
>   - Finally, the local region features from the RoI-Aligned Feature Replay module replace placeholders.
> This strategy allows the LLM's self-attention mechanism to flexibly attend to the global context, the question's semantics, and the specific high-fidelity details of each referenced region, enabling a more powerful and efficient understanding of the scene.

---

### Official Review · Reviewer_uvNs · 2025-11-03

**Soundness:** 4
**Presentation:** 4
**Contribution:** 4
**Rating:** 6
**Confidence:** 4

**Summary:**

This work addresses the limitation of existing MLLMs in fine-grained regional understanding, global context integration, and multi-prompt interaction. It proposes GAR, a model empowered by RoI-aligned feature replay to preserve global context while capturing local details, and GARBench, a benchmark for evaluating single-region comprehension, multi-prompt interaction, and compositional reasoning. GAR outperforms state-of-the-art models on multiple benchmarks and achieves strong zero-shot transfer to video tasks.

**Strengths:**

1. Proposes an innovative RoI-aligned feature replay technique that seamlessly integrates global context and local details, solving a key limitation of existing region-level MLLMs.
2. Develops GARBench, the first benchmark to systematically evaluate multi-prompt interaction and compositional reasoning, filling an evaluation gap.
3. Demonstrates strong generalization, including zero-shot transfer to video tasks, highlighting the model’s practical utility.
Conducts comprehensive experiments (ablation, cross-benchmark, qualitative analysis) that rigorously validate the method’s effectiveness.

**Weaknesses:**

1. RoI-Align’s context binding validity could benefit from additional validation, as the paper does not include targeted experiments for complex scenes where misbinding irrelevant context might occur. Additionally, the ablations in Table 8 do not test whether shielding irrelevant global regions impacts the accuracy of extracted local features.
2. The Grasp Any Region-2.5M dataset and GARBench do not provide clear annotations or statistics on which tasks specifically depend on global context. This may lead to potential over-reliance on global context during training, and it remains unclear if GAR’s performance gains stem from global-local fusion or merely improved local feature capability.
3. GARBench appears statistically limited for drawing robust conclusions, with only 204 samples for GARBench-Cap and 424 samples for GARBench-VQA—sizes that may not fully account for result variability.
4. RoI-Align’s necessity could be further strengthened, as no comparison is provided in the ablation study (e.g., Table 8) with a simpler alternative: directly cropping the local image region and feeding it through the ViT encoder to extract features. It would be helpful to clarify how RoI-Align outperforms this more straightforward approach.

**Questions:**

RoI-Align’s necessity could be further strengthened, as no comparison is provided in the ablation study (e.g., Table 8) with a simpler alternative: directly cropping the local image region and feeding it through the ViT encoder to extract features. It would be helpful to clarify how RoI-Align outperforms this more straightforward approach.

---

> ### Author Response · Authors · 2025-11-25
> **Response to Reviewer uvNs (Part 1)**
>
> We sincerely thank Reviewer uvNs for the thorough review and encouraging feedback. We are extremely pleased with the **"excellent" ratings for our work's soundness, presentation, and contribution**. We are glad the reviewer finds our RoI-aligned feature replay technique "innovative" and recognizes that GARBench is the "first benchmark to systematically evaluate multi-prompt interaction and compositional reasoning", thereby "filling an evaluation gap". We also appreciate the acknowledgment of our "comprehensive experiments" and the model's "strong generalization" and "practical utility."
>
> **W1: About the "misbiding" issue.**
>
> **A1**: We sincerely thank the reviewer for this insightful point. Our core design, which provides the global context as "Image + mask", is precisely intended to mitigate this problem by explicitly guiding the model to focus on the semantically relevant global areas.
>
> To directly address your concern, we have conducted a new, targeted experiment. We created an "**ambiguous context**" setting. In this setting, the global image mask was removed, making global features no longer prompt-aware. Therefore, this design actually simulates "possible misbinding". The local input remained using "RoI-aligned feature replay".
>
> | ID in Table 8 | Global | Local | GARBench-Cap | GARBench-VQA | DLC-Bench |
> |---|---|---|---|---|---|
> | 3 | Image + mask | Image + mask | 28.4 | 36.6 | 77.4 |
> | 4 | Image + mask | RoI-aligned feature replay | 57.5 | 50.6 | 77.1 |
> | Added to 6 | Image | RoI-aligned feature replay | 42.1 | 40.1 | 67.1 |
>
> As shown, not providing global masks, the model's performance on both GARBench-Cap, GARBench-VQA, and DLC-Bench, collapses dramatically, falling back to a level comparable to having no global context at all (Row 3 in Table 8). Adding global prompts naturally solves this "misbinding issue".
>
> *We have added this crucial validation experiment in **Table 5 (the 6th row)** in the revised version. Thank you again for this valuable suggestion.*
>
> **W2: About the annotation or statistics on which tasks specifically depend on global contexts.**
>
> **A2**: First, we provide a detailed explanation of which tasks specifically require global contexts in the following:
> - For the Grasp-Any-Region Dataset-2.5M, 600K relation samples require global contexts explicitly. The rest of 1.9M may include samples that "implicitly" require global contexts, such as the example illustrated in Figure 1a, where precise recognition of the frog-shaped slipper instead of the frog actually requires global contexts.
> - All samples from GARBench-Cap need global contexts, as each of them requires describing the relationship beyond one simple region.
> - Samples from GARBench-VQA-Reasoning require global contexts, since both "position", "non-entity", and "relation" questions require global contexts.
>
> Next, to address the concern of "unclear attribution", we refer to ablation studies in **Table 8**. Experiments in Table 8 actually demonstrate that **the RoI-feature replay technique achieves better global-local fusion while maintaining local details.** Specifically:
> - DLC-Bench is a mainly local-detail benchmark that measures fine-grained local description. Comparing Row 3 (naive global+local cropping) and Row 4 (our GAR with RoI-aligned feature replay), GAR maintains nearly identical local performance. Only a -0.3 drop in DLC-Bench.
> - GARBench (Table 8) is a strict global-context benchmark. Comparing Row 3 and Row 4. GAR achieves massive gains on global-context tasks: +29.1 on GARBench-Cap (28.4 → 57.5) and +14.0 on GARBench-VQA (36.6 → 50.6). These gains are only explainable by better global-local fusion (since local capability is unchanged).
>
> In summary, our GAR manages to understand global contexts (+29.1 on GARBench-Cap and +14 on GARBench-VQA) while maintaining sufficient local details (only -0.3 on DLC-Bench).

---

> ### Author Response · Authors · 2025-11-25
> **Response to Reviewer uvNs (Part 2)**
>
> **W3: About the number of questions of GARBench.**
>
> **A3**: We sincerely thank the reviewer for this important question regarding the robustness of GARBench. To directly address this concern, we conducted a subsampling stability analysis. We randomly subsampled GARBench-VQA and GARBench-Cap to 50% and 25% of their original sizes (for each subtask) and re-evaluated the full suite of models. Our goal was to test if the relative performance rankings remained consistent even with significantly fewer samples.
>
> **The results, presented in the tables below, demonstrate a high degree of ranking stability:**
>
> |  | Full |  | 1/2 |  | 1/4 |  |
> |---|---|---|---|---|---|---|
> | Model | GARBench-VQA | Rank | GARBench-VQA | Rank | GARBench-VQA | Rank |
> | PAM-3B | 2.4 | 9 | 4.3 | 9 | 0.9 | 9 |
> | VP-SPHINX-13B | 37.5 | 8 | 40.0 | 8 | 33.3 | 8 |
> | DAM-3B | 38.2 | 7 | 48.6 | 7 | 41.9 | 7 |
> | GAR-1B | 50.6 | 6 | 51.4 | 6 | 49.5 | 5 |
> | Qwen2.5-VL-32B | 50.9 | 5 | 52.4 | 5 | 48.6 | 6 |
> | GPT-4o | 53.5 | 4 | 56.7 | 4 | 57.1 | 4 |
> | GAR-8B | 59.9 | 3 | 60.0 | 3 | 63.8 | 1 |
> | o3 | 61.3 | 2 | 63.3 | 1 | 58.1 | 3 |
> | Gemini-2.5-Pro | 64.2 | 1 | 61.0 | 2 | 60.0 | 2 |
>
> **The results on GARBench-Cap also demonstrate a high degree of ranking stability:**
>
> |  | Full |  | 1/2 |  | 1/4 |  |
> |---|---|---|---|---|---|---|
> | Model | GARBench-Cap | Rank | GARBench-Cap | Rank | GARBench-Cap | Rank |
> | DAM-3B | 13.1 | 9 | 13.8 | 9 | 14.0 | 9 |
> | PAM-3B | 21.1 | 8 | 18.8 | 8 | 20.1 | 8 |
> | VP-SPHINX-13B | 32.3 | 7 | 29.7 | 7 | 20.0 | 7 |
> | Qwen2.5-VL-32B | 36.8 | 6 | 32.7 | 6 | 26.1 | 6 |
> | GPT-4o | 51.5 | 5 | 45.5 | 5 | 52.1 | 4 |
> | o3 | 56.9 | 4 | 50.6 | 4 | 50.3 | 5 |
> | GAR-1B | 57.5 | 3 | 51.5 | 3 | 53.9 | 3 |
> | Gemini-2.5-Pro | 59.3 | 2 | 54.4 | 2 | 58.1 | 2 |
> | GAR-8B | 62.2 | 1 | 57.4 | 1 | 61.8 | 1 |
>
> As the results show, the relative ordering of models is remarkably stable. For example, in GARBench-Cap, the top 3 models (GAR-8B, Gemini-2.5-Pro, GAR-1B) and bottom 3 models (VP-SPHINX, PAM, DAM) maintain their general ranking group *even at a 1/4 sample size*. While minor fluctuations exist among the top-tier models in the VQA 1/4 split (e.g., GAR-8B jumping from 3rd to 1st), the overall performance tiers are preserved.
>
> This empirical stability aligns with a growing consensus in the evaluation community: **benchmark quality, difficulty, diversity, and representativity, are more critical** than actual sample count. This "less is more" principle is powerfully demonstrated in recent works like LIME [1], TinyBenchmarks [2], and Pacchiardi et al. [3], which show that smaller, carefully curated benchmarks can be more efficient and effective at differentiating models. In designing GARBench, we prioritized this principle, focusing on constructing novel and challenging samples.
>
> *We have added this stability analysis to **Tables 12 and 13** in our revised version.*
>
> **W4: Directly cropping the local image region and feeding it through the ViT encoder to extract features.**
>
> **A4**: We thank the reviewer for this suggestion. While our initial ablation study compared different context combinations (e.g., Row 1st vs. Row 4th in Table 8 in our original manuscript), we agree that this does not fully isolate the contribution of our local feature extraction method against the straightforward baseline you proposed.
>
> To provide a definitive answer, we have followed your advice and implemented this crop baseline (Row 7th). The updated table below presents the direct, head-to-head comparison you requested between our method (Row 4th) and this new baseline (Row 7th)
>
> | ID in Table 8 | Global | Local | GARBench-Cap | GARBench-VQA | DLC-Bench |
> |---|---|---|---|---|---|
> | 1 | -- | Image + mask | 20.1 | 37.8 | 69.3 |
> | 4 | Image + mask | RoI-aligned feature replay | 57.5 | 50.6 | 77.1 |
> | Added to 7 | -- | Image | 17.8 | 35.4 | 65.0 |
>
> The results are decisive. The simple crop-and-resize baseline (Row 7th) performs significantly worse than our full method (Row 4), with performance dropping by -39.7 on GARBench-Cap and -15.2 on GARBench-VQA. This performance gap highlights two fundamental limitations of the simpler approach, which our RoI-aligned feature replay is specifically designed to overcome: (1) loss of high-resolution detail and (2) inability to integrate global context.
> In summary, this new experiment validates that the simple baseline is insufficient, while our method preserves local detail and integrates global context **simultaneously**.
>
> *We have added this crucial comparison to **Table 8** in our revised version.*
>
> [1] LIME: Less Is More for MLLM Evaluation. ACL, 2025.
>
> [2] TinyBenchmarks: evaluating LLMs with fewer examples. ICML, 2024.
>
> [3] 100 instances are all you need: predicting the success of a new LLM on unseen data by testing on a few instances. KDD, 2024.
>
> [4] Not All Data Are Unlearned Equally. COLM, 2025.

---

### Author Response · Authors · 2025-11-25
**General Response and Paper Revision**

We thank all the reviewers for their time, effort, and constructive feedback on our work. We are greatly encouraged by the positive reception and are pleased that the reviewers found our contributions to be significant: (1) there is a consensus that our work addresses an important and challenging problem, and (2) our contributions in the form of the dataset and benchmark are widely appreciated.

We have tried our best to revise the paper to address all concerns. **All revisions are marked in purple.** Specifically:

- In response to **all reviewers**, we have added more ablation studies in **Table 8** in our revised version, including possible "misbinding global context" (Reviewer uvNs), directly cropping the local images (Reviewer uvNs), and GPT4RoI-like architectures (Reviewer n7Lw and ip5b), *where our RoI-aligned feature replay is still the most effective one.*

- In response to **Reviewer ip5b**, we have revised our Introduction, emphasizing *effectively balancing global scene context with fine-grained local details* as our core motivation. Moreover, we have revised our Related Work, providing a more comprehensive and precise categorization of previous approaches.

- In response to **Reviewer SknR**, we have added analysis using different input types in **Table 15**, where powerful general models consistently struggle across all four settings, demonstrating *the fundamental deficiency in fine-grained perception and relational reasoning*, regardless of how the regions of interest are indicated.

- In response to **Reviewer SknR**, we have added more comparisons on general multimodal understanding benchmarks in **Table 11**, where our GAR demonstrates a far superior preservation of general abilities compared to other region-understanding specialist models, and we can substantially recover the general VQA performance by mixing general VQA datasets.

- In response to **Reviewer nvNs** and **Reviewer n7Lw**, we have added robust analysis of GAR-Bench in Tables 12, 13, and 14 in our revised version, where the relative ordering of models is remarkably stable in both 1/2 and 1/4 protocols and different LLM-judges, demonstrating the robustness of our GAR-Bench.

Please let us know if you have any further questions. We are always looking forward to open discussions.
We will give our response as soon as possible once you raise more questions.

Sincerely,

Authors

---

### Meta-Review · Area_Chair_BRWT · 2025-12-15

**Summary:**

* Novelty and Comparison to Prior Art (Reviewers n7Lw, ip5b):
  * Similar to "GPT4RoI" recipe (global image pass + RoI features), and the proposed "RoI-aligned feature replay" seems similar to "RoI-pooling" used in prior papers.
  * Specific comparison baselines (e.g., GPT4RoI-style ablation) were missing.
* Soundness and Baselines (Reviewers uvNs, SknR):
  * The complex architecture might not be necessary, so it’s better to compare with a simple baseline of cropping the local image region.
  * Concerns about "context misbinding": Does the model actually use the global context or just ignore it?
  * The poor performance of baselines might be due to a bias against the "mask" input format rather than a lack of local reasoning capability.
* Evaluation and Benchmarking (Reviewers n7Lw, uvNs, SknR):
  * The proposed GARBench seems too small.
  * Reliance on LLM-based judges (like GPT-4o) was flagged as a potential source of bias.
  * It is better to evaluate on general multimodal benchmarks as well.

**Reviewer Concerns:**

Addressed Concerns:
* Simple Crop Baseline (uvNs): In response to Reviewer uvNs, the authors added a "Simple Crop" baseline.
* Misbinding context (uvNs): The authors added additional ablation to Table 8.
* Global context question (uvNs): The authors explain it using dataset stats and Table 8.
* LLM Judge Bias (n7Lw): A cross-judge consistency table showed stable rankings regardless of the judge used.
* Novelty vs. GPT4RoI (n7Lw, ip5b): The authors performed the exact ablation requested by reviewers, and revised the paper to emphasize the region‑specific reasoning capability as requested by reviewers.
* Comparison with recent region-understanding and grounding works (n7Lw): The authors provided detailed comparison and explanation.
* Mask Bias (SknR): The authors tested SOTA generalized models (e.g., GPT-4o, Gemini) using various input visual prompt types in addition to masks.
* Generalizability (SknR): The authors provided additional results on general benchmarks (e.g., MMVP, RealWorldQA). Original submission also contains several general benchmarks (e.g.,  Ferret-Bench, LVIS, PACO, VideoRefer-Bench, etc.).

Outstanding Concerns:
* Benchmark Stability (uvNs): The authors did a subsampling study showed rankings remained stable even when the benchmark size was reduced by 75%. However, reviewer uvNs is questioning the limited size of GARBench, so it is better to show the robustness by scaling up the dataset to fully address the reviewer’s concern.
* Human Evaluation (n7Lw): Reviewer preferred human ratings over LLM judges. While a cross-LLM validation is reasonable, a large-scale human evaluation would fully address the reviewer’s concern.
* Scalability to the number of regions (n7Lw): The authors did not respond to the concerns about regions scaling.

**Reviewer Scores:**

* Reviewer uvNs
  * Initial rating: 6
  * New rating (potential): 8
  * Reason: The reviewer's request for a "Simple Crop" baseline and "Context Binding" validation were both met.
* Reviewer n7Lw
  * Initial rating: 6
  * New rating (potential): 8
  * Reason: The primary critique was "marginal novelty." The authors' new ablation quantifying the advantage of their architecture over previous works. Another concerns about dependence on LLM judges are mainly addressed by performing cross-judge validation.
* Reviewer ip5b
  * Initial rating: 4
  * New rating (potential): 6
  * Reason: This reviewer explicitly conditioned their assessment of the contribution on the performance of a "GPT4RoI-style ablation." The authors executed this exact request, and the results strongly supported the proposed method.
* Reviewer SknR
  * Initial rating: 4
  * New rating (potential): 6
  * Reason: The reviewer's main skepticism regarding "mask bias" was scientifically refuted by the input format ablation. The generalizability concern was also addressed with new benchmark data.

---

### Decision · Program_Chairs · 2026-01-26

Accept (Poster)